# Insights into rumen microbial biosynthetic gene cluster diversity through genome-resolved metagenomics

Christopher L. Anderson[1] & Samodha C. Fernando [iD] [1✉]

Ruminants are critical to global food security as they transform lignocellulosic biomass into high-quality protein products. The rumen microbes ferment feed to provide necessary energy and nutrients for the ruminant host. However, we still lack insight into the metabolic processes encoded by most rumen microbial populations. In this study, we implemented metagenomic binning approaches to recover 2,809 microbial genomes from cattle, sheep, moose, deer, and bison. By clustering genomes based on average nucleotide identity, we demonstrate approximately one-third of the metagenome-assembled genomes (MAGs) to represent species not present in current reference databases and rumen microbial genome collections. Combining these MAGs with other rumen genomic datasets permitted a phylogenomic characterization of the biosynthetic gene clusters (BGCs) from 8,160 rumen microbial genomes, including the identification of 195 lanthipeptides and 5,346 diverse gene clusters for nonribosomal peptide biosynthesis. A subset of *Prevotella* and *Selenomonas* BGCs had higher expression in steers with lower feed efficiency. Moreover, the microdiversity of BGCs was fairly constant across types of BGCs and cattle breeds. The reconstructed genomes expand the genomic representation of rumen microbial lineages, improve the annotation of multi-omics data, and link microbial populations to the production of secondary metabolites that may constitute a source of natural products for manipulating rumen fermentation.

[1] Department of Animal Science, University of Nebraska, Lincoln, NE, USA. ✉email: samodha@unl.edu

With the expected population growth and changes in food consumption patterns, ruminant agriculture is critical to meeting global demands for animal products[1]. The rumen microbial community is central to the conversion of indigestible plant biomass into food products via the breakdown of complex carbohydrates to volatile fatty acids that provides the ruminant animal with ~70% of its caloric requirements[2]. Consequently, rumen microbes are paramount to ruminant health and productivity. Advancing the understanding of the structure–function relationship of the rumen microbiome is critical to improving ruminant agriculture.

Recent investigations of the rumen microbiome have expanded rumen microbial genomic databases[3–5]; however, the genomic characterization of rumen microbes is far from complete. The Hungate1000 project provided high-quality reference genomes for hundreds of cultured rumen microbial strains[3]. New culturing efforts and strategies are bound to bring more rumen microbes into culture, but currently, the majority of populations have not been isolated. In a recent cultivation experiment with defined and undefined media, 23% of rumen microbial operational taxonomic units were recovered[6]. Metagenomic binning approaches have been employed to bypass the cultivation bottleneck and generate rumen microbial population genomes[4,5,7,8]. Stewart et al. reconstructed 4,941 metagenome-assembled genomes (MAGs) from cattle and highlighted the carbohydrate-active enzyme diversity residing in uncultivated taxa[4,5]. However, a notable fraction of metagenomic reads from previous studies did not map to genomes from the Stewart et al. and Hungate1000 collections[4,5], suggesting many rumen microbial species are yet to be characterized. Increasing the number of reference genomes for rumen microbes by identifying MAGs across different ruminant species would enhance our understanding of structure–function relationships within the rumen microbiome and improve metagenomic inference.

Secondary metabolites are involved in a broad range of functions, including as antimicrobial agents and mediating microbial interactions[9]. Given the evidence linking the transmission of antibiotic resistance from livestock to humans[10,11], there is a need to reduce the use of antimicrobial feed additives by identifying alternatives[12,13]. Due to the intense competition for nutrient resources, the rumen microbiome may provide novel opportunities to develop alternatives using endogenous antimicrobial peptides and probiotic microbial species[14]. A previous analysis found 45.4% of 229 rumen genomes encoded at least one bacteriocin gene cluster[15]. Secondary metabolites also have ecological roles in intercellular communication[9,16]. In support, a recent study of rumen metatranscriptomic data demonstrated increased expression of nonribosomal peptide (NRPS) and polyketide synthetases (PKS) in some rumen species during colonization of plant biomass, suggesting roles for these molecules in establishing niches in the rumen[17]. Thus, expanding on these findings and gaining additional fundamental knowledge on microbial secondary metabolism in the rumen is important for understanding host–microbe and microbe–microbe interactions, as well as for developing alternative compounds to improve ruminant health and modulate rumen fermentation.

Here, we used publicly available metagenomes from ruminants (cattle, deer, moose, bison, and sheep) in combination with new cattle rumen metagenomic datasets to reconstruct 2,809 MAGs. The MAGs expand the genomic representation of rumen microbial lineages and provide unique genomic insights into rumen microbial physiology. Moreover, we present a phylogenetic characterization of the secondary metabolite biosynthetic gene clusters (BGCs) of rumen microbial genomes and demonstrate the vast potential present within the rumen microbiome for the discovery of novel metabolites and probiotics to improve animal health and productivity.

## Results

**2,809 draft MAGs from the rumen ecosystem.** We amassed 3.2 terabase pairs (Tbp) of data from 346 publicly available and 66 new rumen metagenome datasets (Supplementary Table 1). The metagenomes were from cattle (312 samples, 2.1 Tbp), sheep (75 samples, 888.4 gigabase pairs (Gbp)), moose (9 samples, 108.8 Gbp), deer (8 samples, 62.9 Gbp), and bison (8 samples, 52.3 Gbp). Metagenomes were assembled independently to reduce the influence of strain variation and improve the recovery of closely related genomes[18,19]. Following refinement, dereplication, and filtering of resulting population genomes, we identified 2,809 nonredundant MAGs satisfying the following criteria: dRep[20] genome quality score ≥60, ≥75% complete, ≤10% contamination, N50 ≥5 kbp, and ≤500 contigs.

The median estimated completeness and contamination of the MAGs were 89.7% and 0.9%, respectively (Fig. 1a and Supplementary Data 1). Further, recovered MAGs had a median genome size of 2.2 Mbp, a median of 131 contigs, and a median N50 of 28.3 kbp (Fig. 1b). The proposed minimum information about a MAG (MIMAG) specifies high-quality draft genomes to have an estimated ≥90% completeness, ≤5% contamination, at least 18 tRNAs, and contain 23S, 16S, and 5S rRNA genes[21]. It remains challenging to reconstruct rRNA genes from short metagenomic reads due to the high sequence similarity of rRNA genes in closely related species. As a result, despite high estimated completeness and low contamination rates, only 20 MAGs meet the MIMAG standards for a high-quality draft genome. We identified a 16S rRNA gene in 197 of the MAGs. The remaining MAGs are characterized as medium-quality MAGs under the MIMAG standards.

The majority of bacterial MAGs belonged to phyla Firmicutes or Bacteroidota (2,326; Fig. 2a and Supplementary Data 1). Additionally, we assembled 12 bacterial genomes from the superphylum Patescibacteria. At lower taxonomic ranks, Lachnospiraceae (415) and *Prevotella* (398) were the dominant family and genus identified among the assembled bacterial genomes. The most prevalent archaeal family and genus were Methanobacteriaceae (45) and *Methanobrevibacter* (35), respectively (Fig. 2b). The recovered MAGs represent several new taxonomic lineages, as four genomes could not be classified at the rank of order, 16 at the rank of family, and 243 at the genus rank.

**Species-level overlap between reference genomes, the Hungate1000 Collection, and rumen MAGs.** To further characterize the assembled genomes, we compared the MAGs to other rumen-specific genome collections, specifically genomes generated from the Hungate1000 project[3] and MAGs identified from the Stewart et al. studies[4,5]. We clustered genomes based on approximate species-level thresholds (≥95% ANI) and calculated the intersection between MAGs in the current study and the Hungate1000 Collection (410 genomes)[3], MAGs from Stewart et al. (4,941 genomes)[4,5], and a dereplicated genome collection from the GTDB (22,441 genomes, see Methods)[22], which includes reference isolate genomes and some environmental MAGs[23]. It should be noted that we used the raw data from the first of the Stewart et al. studies[4] (Supplementary Table 1), but with different assembly and binning approaches. Approximately one-third of the MAGs (1,007) did not exhibit ≥95% ANI with a genome in the GTDB, Stewart et al. MAGs, or the Hungate1000 isolates (Fig. 3a). When considering the pairwise intersections between the datasets, 98 (3.5%), 933 (33.2%), and 1,438 (51.2%) of the MAGs in the current study had ≥95% ANI with a genome in the Hungate1000 Collection[3], GTDB[22], and Stewart et al.[4,5], respectively. One hundred twenty-one (29.5%), 552 (2.5%), and 3,125 (63.2%) of the genomes from the Hungate1000 Collection[3],

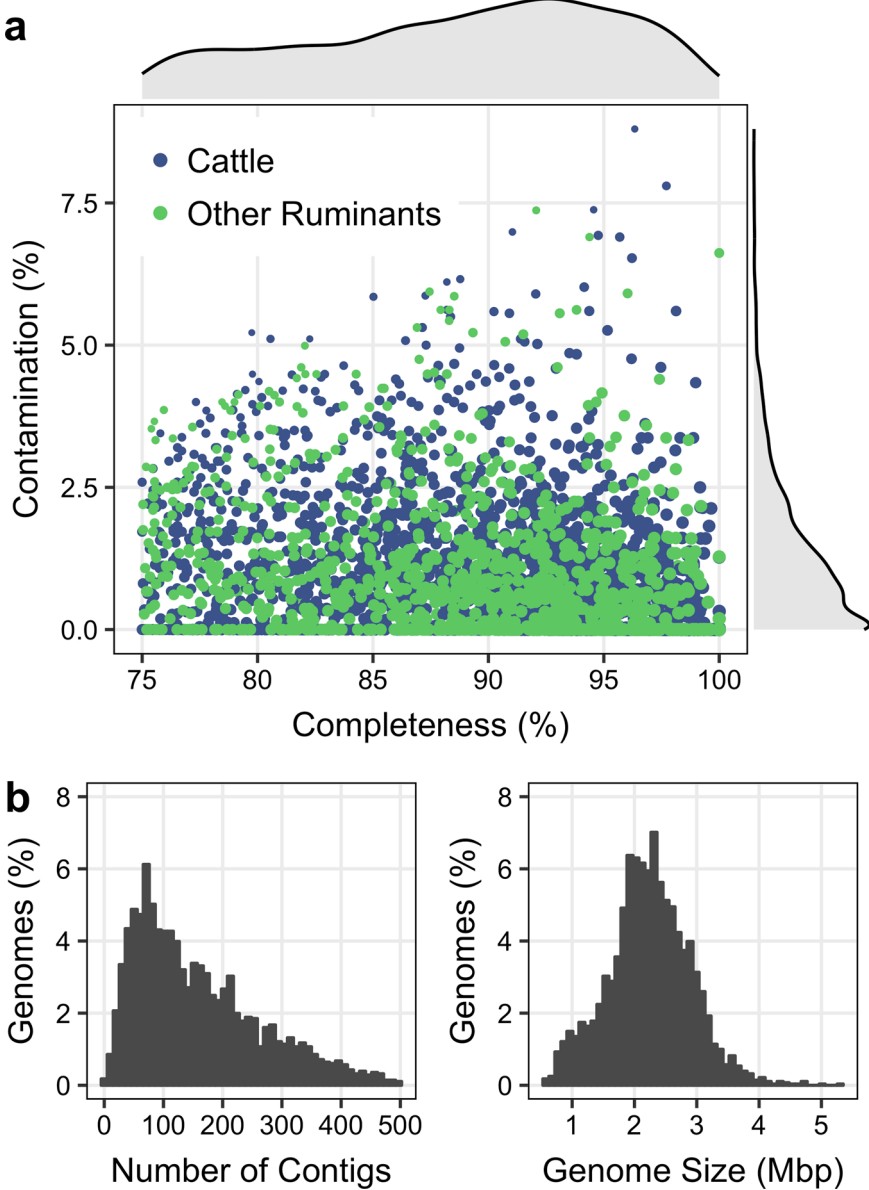

**Fig. 1 Genomic properties of 2,809 rumen MAGs. a** CheckM completeness and contamination estimates for the 2,809 population genomes recovered from rumen metagenomes. The size of the point on the scatter plot corresponds to the dRep genome quality score, where Quality = Completeness − (5 · Contamination) + (Contamination · (Strain Heterogeneity/100)) + 0.5 · (log (N50). The reported MAGs meet the following minimum criteria: genome quality score ≥60, ≥75% complete, ≤10% contamination, N50 ≥5 kbp, and ≥500 contigs. **b** The frequency distribution of the number of contigs and genome sizes of reconstructed MAGs.

GTDB[22], and Stewart et al.[4,5] displayed ≥95% ANI with a MAG from the current study. Together, these results indicate that we recovered a majority of previous rumen genomic diversity with additional lineages not previously identified in other major rumen genomic collections.

We applied an additional clustering approach to identify the approximate number of species represented by the rumen-specific genomes assembled in this study, in the Hungate1000 Collection[3], and Stewart et al.[4,5]. A 95% ANI threshold yielded 3,541 clusters from the combination of the datasets (Supplementary Data 2). Of the 3,541 clusters, 2,024 contained a MAG from the current study, and 1,135 were composed exclusively of MAGs from the current study. In comparison, 2,175 and 286 clusters were comprised of genomes from Stewart et al.[4,5] and the Hungate1000 Collection[3], respectively. The majority of 95% ANI clusters (2,166) are only comprised of a single genome (Fig. 3b).

Furthermore, a rarefaction curve suggests the 8,160 genomes from the genomic collections analyzed here only represent a fraction of the estimated microbial species diversity in the rumen (Fig. 3c). The genome with the best dRep score from each cluster was used to generate a phylogenetic tree highlighting the species diversity within each rumen genomic collection and represents the vast diversity of rumen bacterial (Fig. 3d) and archaeal (Fig. 3e) genomes published to date.

As stated previously, the median genome size of reconstructed MAGs was 2.2 Mbp, smaller than the median size of genomes from the Hungate1000 project (3.1 Mbp)[3]. To provide an assessment at a finer resolution, genome sizes of MAGs and Hungate1000 genomes[3] belonging to the same 95% ANI cluster were compared (Supplementary Fig. 1). Adjusted sizes of MAGs and Hungate1000 genomes that are ≥95% complete displayed a regression coefficient of 0.96 with a slope of 0.86, indicating the

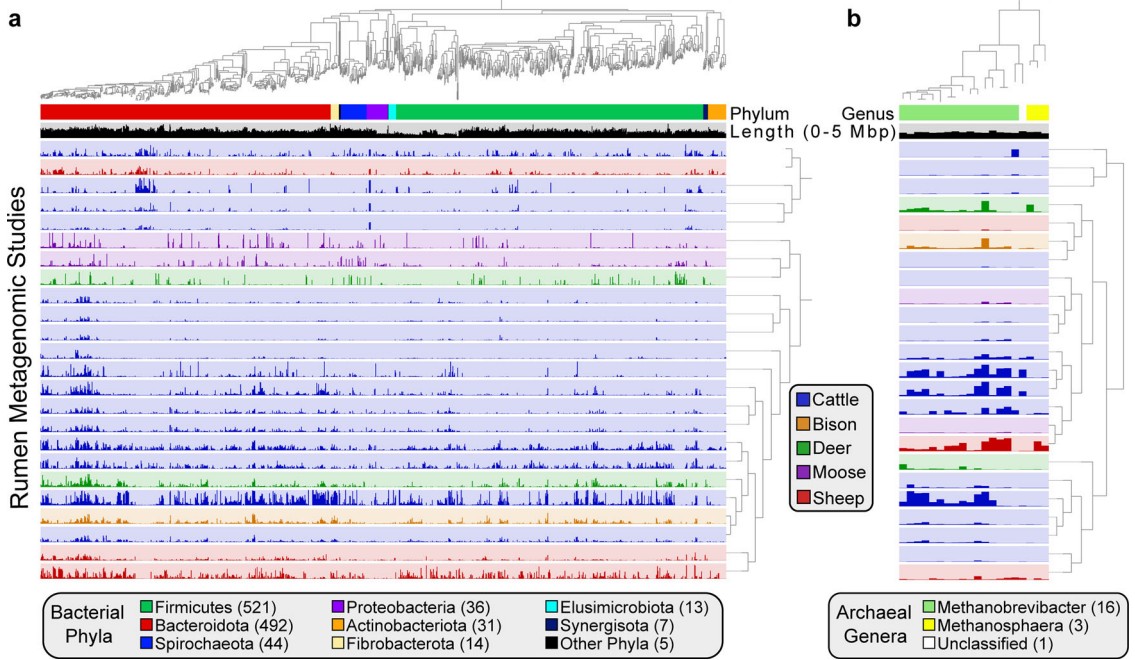

**Fig. 2 Phylogenetic relationships and coverage patterns of near-complete MAGs. a** Phylogenomic analysis of 1,163 near-complete (≥90% complete, ≤5% contamination, and N50 ≥15 kbp) bacterial MAGs and (**b**) 20 near-complete archaeal MAGs inferred from the concatenation of phylogenetically informative proteins. Layers below the genomic trees designate bacterial phylum or archaeal genus based on GTDB taxonomic assignments, genomic size (0–5 Mbp), and the mean number of bases with ≥1× coverage in a rumen metagenomic dataset (layer color indicates the ruminant the data was collected from). The mean number of bases with ≥1× coverage was used as input for hierarchical clustering of rumen metagenomic datasets based on Euclidean distance and Ward linkage. The bacterial and archaeal phylogenetic trees are provided as Supplementary Data 6 and Supplementary Data 7, respectively.

binning process likely did not lead to extensive losses and systematic biases in the reconstructed genomes. Instead, it further highlights that current culturing approaches have not brought large portions of rumen microbial diversity into culture and putatively supports previous findings from the human gut that revealed genome-reduction in uncultured bacteria[24].

**Rumen metagenome classification rates using reference and rumen-specific genomes**. Utilizing an approach similar to Stewart et al.[4,5], we investigated the influence of MAGs on rates of metagenomic read classification. The baseline for read classification was the standard Kraken database containing bacterial, archaeal, fungal, and protozoal RefSeq genomes[25]. Each rumen-specific dataset was incrementally added to the Kraken RefSeq genomic database in the following order to build new databases: the Hungate1000 Collection[3], MAGs from Stewart et al.[4,5], and MAGs from the current study. Each individual and collective database was used for classification of sample reads that underpinned metagenomic binning and from a rumen metagenomic dataset not used in the reconstruction of MAGs[26]. MAGs from the current work classified more reads from deer, moose, and sheep metagenomes, while the more numerous MAGs from Stewart et al.[4,5] classified more reads from bison and cattle metagenomes (Supplementary Fig. 2a). The addition of MAGs improves classification relative to databases primarily based on cultured isolates, like the Hungate1000 Collection[3] (Supplementary Fig. 2b). Using the combination of all reference and rumen-specific genomes, the median classification rate on an independent set of cattle metagenomes was 62.6%.

**Phylogenetic characterization of biosynthetic gene clusters**. Microbial genome mining is a powerful tool for natural product discovery. We sought to explore the extent of secondary

metabolite diversity coded by the MAGs in the current study, the Hungate1000 Collection[3], and Stewart et al. MAGs[4,5]. We identified 14,814 BGCs encoded by the 8,160 rumen-specific genomes using antiSMASH[27] (Fig. 4a and Supplementary Data 3). The majority of BGCs were NRPS (5,346), followed by aryl polyenes (2,800), sactipeptides (2,126), and bacteriocins (1,943). Only a few PKS were identified (75). Firmicutes harbored the vast majority of clusters for NRPS, sactipeptide, lantipeptide, lassopeptide, and bacteriocin synthesis (Fig. 4b). At lower taxonomic ranks, DTU089 (979), Bacteroidaceae (934), and Lachnospiraceae (923) coded for the bulk of NRPS gene clusters. Moreover, Acid-aminococcaceae genomes contained 21.2% of identified bacteriocins and *Ruminococcus spp.* possessed the bulk of sactipeptides and lantipeptides. Archaea were predicted to code 737 BGCs, including an average of 3.8 NRPS gene clusters per genome (Fig. 4a).

NRPS exhibit high molecular and structural diversity resulting in a wide array of biological activities. The diversity of NRPS, combined with their proteolytic stability and selective bioactivity, has resulted in the development of many NRPS as antimicrobials and other therapeutic agents[28]. Given the prevalence of NRPS among the recovered MAGs (Fig. 4a), the peptides appear to be important bioactive metabolites in the rumen. To gain fundamental insight into the phylogenetic diversity of rumen NRPS, we built a network based on BGC similarity using BiG-SCAPE[29]. BiG-SCAPE uses protein domain content, order, copy number, and sequence identity to calculate a distance metric. We assessed the similarity of NRPS gene clusters identified in Firmicutes, Bacteroidota, and Euryarchaeota, as these three phyla coded for 96.4% of assembled NRPS gene clusters from rumen genomes. With a BiG-SCAPE similarity threshold of 0.3, the resulting network consisted of 3,436 nodes (NRPS BGCs on contigs ≥10 kbp) and 79,112 edges (Fig. 4c and Supplementary Data 4). As expected, the network analysis depicted high inter- and

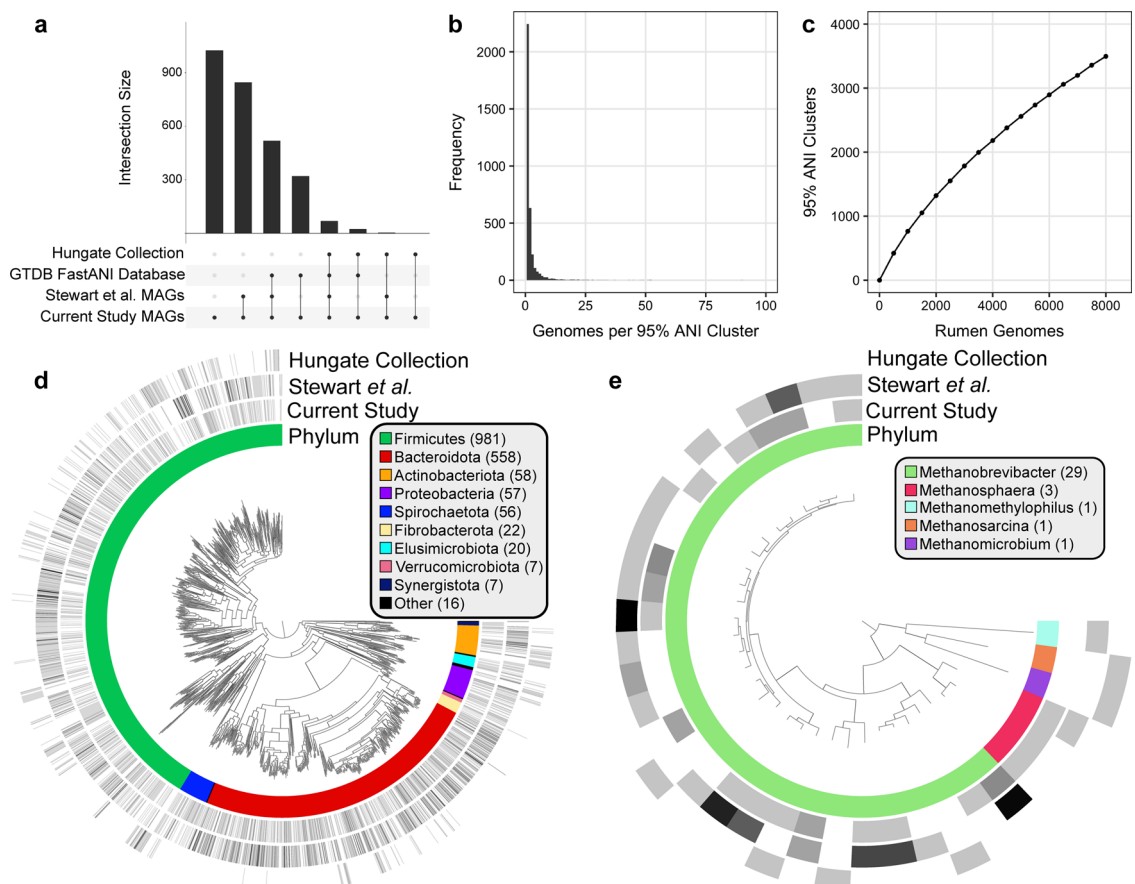

**Fig. 3 Genomes sharing ≥95% ANI between databases and the characterization of rumen-specific 95% ANI clusters. a** The approximate number of species overlapping amongst rumen-specific and reference genomic datasets. Genomes demonstrating ≥95% ANI were considered to be shared between two datasets. Presented are a subset of intersections in which a MAG from the current study was the query genome. **b** The number of genomes comprising each of the 3,541 95% ANI clusters generated from 8,160 rumen microbial genomes in the current study, the Hungate1000 Collection[3], and Stewart et al. studies[4, 5]. **c** Rarefaction analysis based on subsampling 95% ANI clusters at steps of 500 genomes indicates the 8,160 genomes from recently published rumen genomic collections still only represent a fraction of expected microbial species diversity in the rumen ecosystem. Phylogenomic relationships of the 1,781 near-complete bacterial (**d**) and 35 near-complete archaeal (**e**) representative genomes with the highest dRep genome quality score from the 3,541 95% ANI clusters generated from 8,160 rumen-specific genomes. Near-complete genomes were defined as being ≥90% complete, having ≤5% contamination, and contig N50 ≥15 kbp. Layers surrounding the genomic trees indicate the bacterial phyla or archaeal genera and the log normalized number of genomes from each rumen genomic collection belonging to the same 95% ANI cluster. The bacterial and archaeal phylogenetic trees are provided as Supplementary Data 8 and Supplementary Data 9, respectively.

intra-phylum genetic diversity among the NRPS gene clusters. The median intra-phylum, -family, and -genus similarity was 0.40, 0.44, and 0.46, respectively, while the median inter-phylum, -family, and -genus similarity was 0.32, 0.34, and 0.34, respectively. Further, only 2.6% of edges were inter-phylum and 69.0% were intra-family. Of the 6,594 Euryarchaeota edges, 8.1% were Euryarchaeota-Firmicutes (median similarity of 0.32) and 2.0% of edges were Euryarchaeota-Bacteroidota (median similarity of 0.31). To further examine the phylogenetic relationships of rumen Euryarchaeota NRPS, we clustered 265 NRPS gene clusters (≥10 kbp) from 85 near-complete Euryarchaeota genomes at a higher similarity threshold of 0.75, yielding 57 NRPS clusters (Fig. 4d). The distribution of NRPS clusters amongst the genomes suggests there exists a strong relationship between methanogen phylogeny and NRPS similarity. Only *Methanobrevibacter* genomes contain NRPS gene clusters, and genomes of the same species often possessed many of the same NRPS clusters (see genomes highlighted in blue in Fig. 4d). However, there are instances in which closely related methanogens code for a contrasting pattern of NRPS clusters or no NRPS clusters at all (see genomes highlighted in red in Fig. 4d).

Bacteriocins likely serve as regulatory elements in complex microbial communities such as the rumen. Consequently, bacteriocins have been studied and characterized for their bactericidal activity and as agents that modulate the microbiota structure and function[30]. In particular, lanthipeptides, a class of ribosomally synthesized and post-translationally modified peptides (RiPPs) with thioether cross-linked amino acids[31], are of pharmaceutical, preservative, and agricultural interest due to their strong antimicrobial properties against gram-positive pathogens[31–33], low levels of antimicrobial resistance[34], and stability[35]. We identified 195 rumen lanthipeptide BGCs from the Hungate1000 genomes and MAGs from Stewart et al. and the current study. Rumen lanthipeptide BGCs were clustered with 22,870 lanthipeptide BGCs from RefSeq genomes[36,37] into gene cluster families (GCFs; groups of BGCs that may generate highly similar products). Clustering with BiG-SCAPE[29] yielded 4,565 GCFs, 120 of which contained a rumen lanthipeptide. The 120 GCFs were composed of 519 lanthipeptide BGCs, where 324 were from RefSeq isolates and 195 from rumen genomes (Fig. 5a). The 324 RefSeq BGCs fell into only 18 GCFs. Lanthipeptides from the Hungate1000 isolates clustered into 36 GCFs, while rumen MAG

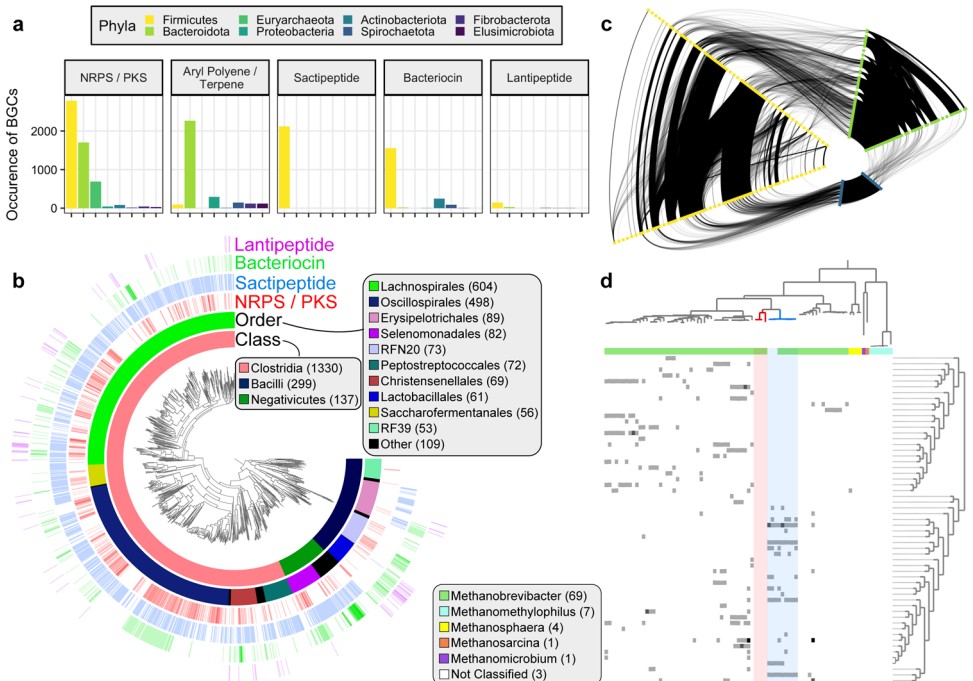

**Fig. 4 Characterization of BGCs from 8,160 rumen genomes and MAGs. a** Number and types of BGCs identified from select phyla in genomes from the Hungate1000 Collection[3], Stewart et al. studies[4, 5], and the current study. **b** Phylogenomic analysis of 1,766 near-complete Firmicutes genomes inferred from the concatenation of phylogenetically informative proteins. The inner layer surrounding the genomic tree designates taxonomic annotations, while the remaining layers depict the log normalized number of BGCs in the genome with the ascribed function. Bacterial class and order labels are displayed for those lineages in which more than 50 genomes were identified. Near-complete genomes were defined as being ≥90% complete, having ≤5% contamination, and contig N50 ≥15 kbp. The phylogenetic tree is provided as Supplementary Data 10. **c** A relational network of NRPS gene clusters in Firmicutes, Bacteroidota, and Euryarchaeota highlights the similarity of NRPS BGCs from Euryarchaeota and Firmicutes. Edge weight represents the similarity of two BGCs, as determined by BiG-SCAPE (i.e. darker edges demonstrate more similarity between two BGCs). Edges are only shown for BGCs with ≥0.3 BiG-SCAPE similarity. Nodes from each phylum are duplicated to illustrate intra-phylum relationships and nodes along a given axis are ordered alphabetically by taxonomic family. **d** The association between genome phylogeny and the similarity of NRPS gene clusters coded by near-complete Euryarchaeota genomes. BGCs designated as NRPS were clustered with BiG-SCAPE. The relationship between NRPS clusters was portrayed through the hierarchical clustering of pairwise inter-cluster similarities. The number of NRPS clusters coded by each genome (range of 0–3) is presented alongside the assigned genus. A group of *Methanobrevibacter* genomes, likely of the same species (≥95% ANI), possessed very similar NRPS clusters (highlighted in blue). Yet, phylogenetically closely related genomes, belonging to two different 95% ANI clusters, did not code for any identified NRPS gene clusters (highlighted in red). The phylogenetic tree is based on the concatenation of 122 phylogenetically informative archaeal proteins and is available as Supplementary Data 11.

lanthipeptides belonged to 92 GCFs, 82 of which were exclusively composed of MAG lanthipeptides. Together, this evidence suggests rumen MAGs code for diverse and novel lanthipeptides not represented in cultured isolates, including the Hungate Collection.

We sought to further examine the differences in rumen MAG lanthipeptides relative to isolates and the taxonomic diversity of rumen microbes coding for lanthipeptides. The 195 rumen lanthipeptides were mainly found in Firmicutes genomes, with a subset from Bacteroidota and Actinobacteriota (Fig. 5b). Fifty-two of the 55 lanthipeptides from the Hungate Collection isolates were from Firmicutes (94.5%). At the family-level, these 52 Firmicutes BGCs were distributed evenly between Lachnospiraceae and Streptococcaceae. In contrast, 19.2% and 8.6% of lanthipeptides from rumen MAGs belonged to Bacteroidota and Actinobacteriota, respectively. Lanthipeptides from MAGs were also found in Muribaculaceae and Oscillospiraceae. Moreover, 26.4% of rumen MAG lanthipeptides, compared to 3.6% of Hungate Collection isolates, were found in *Eubacterium* genomes. The majority of *Eubacterium* MAG lanthipeptides (62.1%) belonged to a single GCF, suggesting they code for very similar products. Lastly, antiSMASH predicted the bulk of the rumen lanthipeptides were Class II lanthipeptides, with fewer Class I and

Class III types (Fig. 5b). Nearly all of the Class I lanthipeptides were from Hungate isolates. The above analysis of lanthipeptide diversity further supports that rumen MAGs code for novel secondary metabolites not represented in cultured isolates.

We aligned previously published rumen metatranscriptome data from steers characterized as having high and low feed efficiency to the BGCs to demonstrate if the identified BGCs are active and to explore potential ecological roles of secondary metabolites. Despite data from the metatranscriptome study not being applied to reconstruct genomes in the current study, we identified the expression of 554 gene clusters from rumen-specific genomes in the 20 metatranscriptomes (≥100 aligned reads). Metatranscriptome read count data were normalized independently for each genome to better account for the variation in taxonomic composition across samples[38]. Genome-specific normalization resulted in the identification of 17 differentially expressed gene clusters between steers with high and low feed efficiency (DESeq2[39] false discovery rate adjusted $P < 0.05$; Supplementary Data 5). Of the 17 differentially expressed BGCs, 16 exhibited higher expression levels in the rumen samples from less efficient steers with higher residual feed intake. Further, *Prevotella* and *Selenomonas* coded for 12 of the differentially expressed BGCs (70.6%). All of the differentially expressed

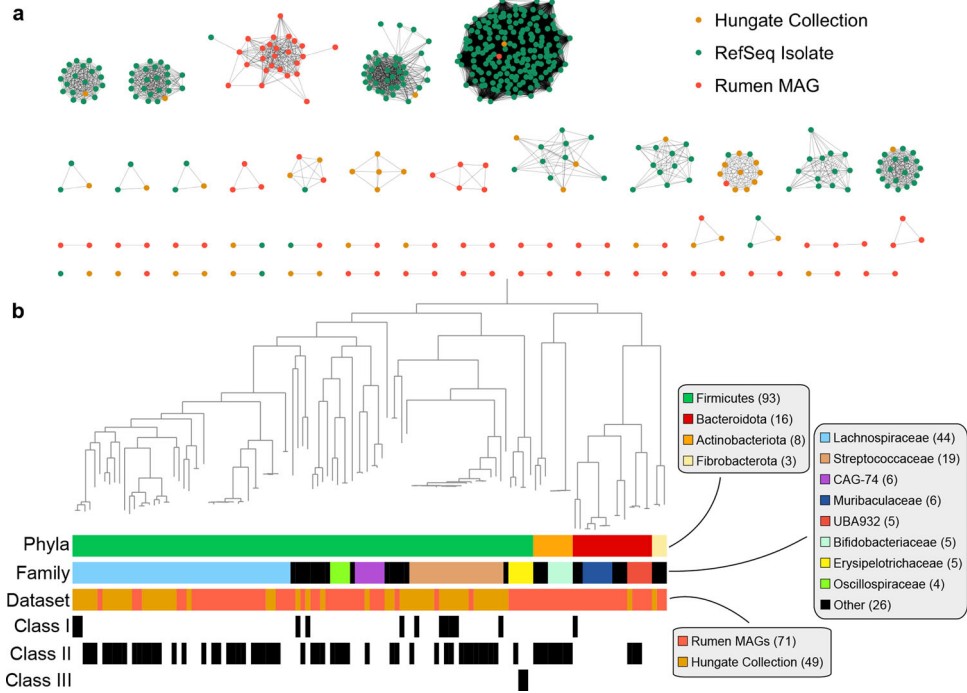

**Fig. 5 Phylogenetic diversity of 195 lanthipeptide BGCs coded by rumen genomes. a** Network depicting the similarity between lanthipeptide BGCs identified from complete and draft isolate genomes in RefSeq and rumen genomes of the Hungate1000 collection, Stewart et al. MAGs, and MAGs from the current study. The BGCs were clustered into gene cluster families (GCFs) with BiG-SCAPE[29]. Only the GCFs containing a rumen genome and at least two BGCs were visualized. Nodes in the network represent BGCs and edges connect BGCs with BiG-SCAPE defined similarity ≥0.3. **b** Phylogenetic relationships of 120 near-complete rumen bacterial genomes coding for lanthipeptide BGCs. Near-complete genomes were defined as being ≥90% complete, having ≤5% contamination, and contig N50 ≥15 kbp. Layers surrounding the genomic trees indicate the bacterial phyla and family, if the genome is a MAG or Hungate Collection isolate, and the class of lanthipeptide, as predicted by antiSMASH[27]. Genomes without an indicated lanthipeptide class were not classified by antiSMASH. The phylogenetic tree is based on the concatenation of 120 phylogenetically informative bacterial proteins and is available as Supplementary Data 12.

*Selenomonas* BGCs were sactipeptides ($n = 7$), while the *Prevotella* BGCs were more diverse and included NRPS and aryl polyenes.

**Microdiversity of BGCs and MAGs**. Phylogenetic analyses of BGC often revealed high inter-species diversity (i.e, methanogen NRPS diversity in Fig. 4d). We next investigated patterns of sub-species microdiversity in rumen BGCs. In order to reduce the influence of study-to-study effects, we focused on the microdiversity of MAGs across 282 metagenomes in the Stewart et al. studies[4,5]. MAGs with ≥50% of its genome covered by at least 5 reads were considered as detected in a sample and used for microdiversity analyses. The within-sample microdiversity of genes and genomes were assessed using InStrain[40]. Our phylogenetic analysis identified that different classes of BGCs are enriched in certain lineages (Fig. 4a, b). As a result, the nucleotide diversity values for genes were normalized using the mean genome-wide nucleotide diversity for each MAG to account for lineage-specific evolutionary processes and more accurately compare patterns of microdiversity in BGCs across lineages. There were significant differences in the nucleotide diversity of genes from the four major classes of BGCs identified in rumen-specific genomes (Kruskal–Wallis $H = 1795.5$, $\varepsilon^2 = 0.001$, $P < 2.2 \times 10^{-16}$; Fig. 6a), but the effect size ($\varepsilon^2$) between BGC types was negligible. Outliers with high microdiversity were bacteriocin genes from *RC9* and *UBA3207 sp.* as well as NRPS genes from *CAG-710* and *UBA9715 sp.* Additionally, we explored the association of genome-wide and secondary metabolism gene microdiversity with cattle breed. The mean nucleotide diversity of MAGs (Kruskal–Wallis $H = 1027.5$, $\varepsilon^2 = 0.0265$, $P < 2.2 \times 10^{-16}$;

Fig. 6b) and the normalized nucleotide diversity of genes from BGCs (Kruskal–Wallis $H = 403.84$, $\varepsilon^2 = 0.0003$, $P < 2.2 \times 10^{-16}$; Fig. 6c) were both significantly different between the four breeds. The effect size ($\varepsilon^2$) of microdiversity difference between breeds was much larger for the genome-wide comparison than for genes from BGCs. This finding raised the question if genes from BGCs have different nucleotide diversity relative to other genes. We found that genes across all BGCs had lower normalized nucleotide diversity compared to all other genes from investigated MAGs (Wilcoxon rank-sum $W = 6.11 \times 10^{13}$, Vargha and Delaney's $A = 0.507$, $P < 2.2 \times 10^{-16}$; Fig. 6d). The raw nucleotide diversity values were higher for genes in BGCs than other genes (Wilcoxon rank-sum $W = 5.801 \times 10^{13}$, Vargha and Delaney's $A = 0.481$, $P < 2.2 \times 10^{-16}$). Regardless, again we find the effect size of the difference to be very small though. Together, microdiversity analyses suggest rumen microbial BGC diversity is comparable across the prevalent BGC classes, breeds, and similar to other genes.

## Discussion

Ruminant agriculture is critical to the global food system. However, with land constraints and associated environmental impacts, ruminant production systems will need to become more efficient and sustainable to feed a growing population. Due to the importance of microbial processes in ruminant health and production, rumen microbes are central to nearly all aspects of ruminant agriculture[41]. Actionable insights into the roles of rumen microbes have lagged though, partly due to a lack of genomic references that underpin analyses and contextualize community data.

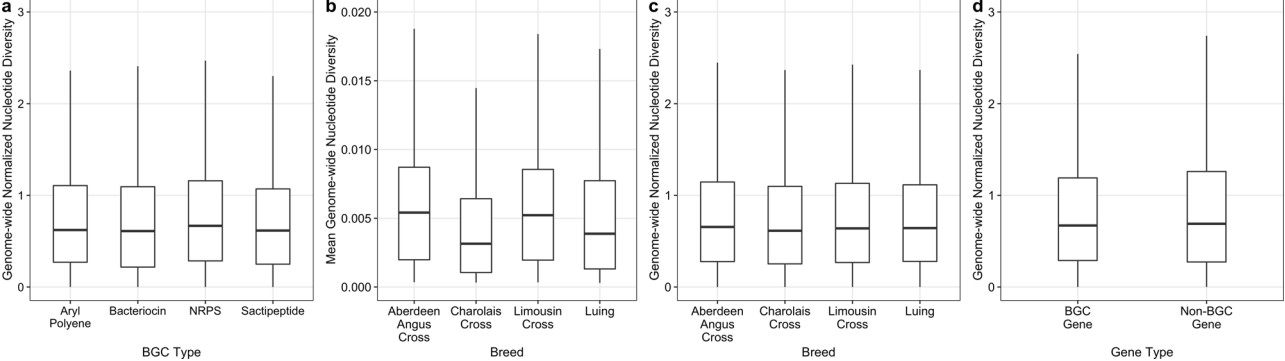

**Fig. 6 Comparison of the microdiversity of MAGs and BGCs from cattle metagenomes in the Stewart et al. studies[4, 5].** The within-sample nucleotide diversity of BGCs was statistically different between BGC types, but the effect size of the difference was small ($\varepsilon^2 = 0.001$) (**a**). The difference in nucleotide diversity across breeds was greater for MAGs ($\varepsilon^2 = 0.0265$) (**b**) than for genes in BGCs ($\varepsilon^2 = 0.0003$) (**c**). Additionally, the effect size of the difference between the normalized nucleotide diversity of genes from BGCs and other genes was small (Vargha and Delaney's $A = 0.507$) (**d**). Genome-wide normalized nucleotide diversity is the nucleotide diversity of a gene relative to the mean nucleotide diversity of the MAG. The genome-wide normalized nucleotide diversity metric was used to reduce the influence of lineage-specific evolutionary processes, allowing for a more accurate comparison of gene nucleotide diversity across microbial populations. The same conclusions were identified using the raw nucleotide diversity in the place of genome-wide normalized nucleotide diversity. The outlier points have been removed from the boxplots for clarity.

We reconstructed 2,809 metagenome-assembled population genomes from several ruminant species to advance our understanding of the structure–function relationship of the rumen microbial ecosystem. Nearly half of the MAGs are estimated to be ≥90% complete with minimal contamination. Based on pairwise ANI comparisons, the MAGs in this study constitute ~2,024 species (95% ANI clusters), greatly expanding the genomic representation of rumen microbial lineages. Moreover, clustering the genomic data reported in this study with genomes from the Hungate1000 Collection[3] and Stewart et al. studies[4,5] suggest there are at least 3,541 rumen microbial species with a draft reference genome now. It is worth emphasizing that some of the MAGs reported in this study may have been reported as part of metagenomic binning efforts in other recent studies[42−44] (Supplementary Table 1). The aggregating of data from multiple studies and contrasting assembly and binning approaches may have recovered different microbial populations. In particular, the pooling and binning of contigs from an increased number of samples followed by re-assembly likely yielded more MAGs, MAGs of improved quality, and the recovery of different genetic elements.

Approximately one-third of the resolved MAGs did not have a species-level representative in the compared genomic databases. Among the fraction of genomes that did exhibit high similarity, only 3.7% of MAGs formed a cluster with 29.8% of genomes in the Hungate1000 Collection[3]. Further, 64.6% of the Hungate1000 Collection genomes did not cluster with a MAG from the current study or the Stewart et al. studies[4,5], implying metagenomic binning did not recover some of the cultured rumen isolates. The poor reconstruction of isolate genomes may be because the Hungate1000 strains are in low abundance in vivo or have high intra-species diversity. An examination of culturing with defined and undefined media found that a large number of cultured OTUs were not detected in the 16S rRNA gene profile from the same rumen sample or were unique to a culture plate, suggesting cultured OTUs often constituted rare rumen microbial populations[6]. The addition of rumen MAGs to classification indices may improve statistical power and allow for a more accurate interpretation of shallow rumen metagenomic datasets[45]. Therefore, the MAGs presented here are valuable for interpreting future and previously sequenced rumen metagenomic datasets and serving as a scaffold for other multi-omics data.

Moreover, we linked microbial populations to the coding and expression of BGCs to demonstrate the utility of genome-resolved metagenomics in the rumen ecosystem. This analysis identified 14,814 gene clusters from 8,160 rumen-specific genomes, indicating the rumen is a rich resource for secondary metabolites. Previous investigations of rumen secondary metabolites have primarily focused on bacteriocin production. A similar genome mining approach revealed 46 bacteriocin gene clusters from 33 rumen bacterial strains[15]. Roughly half the clusters were related to lanthipeptide biosynthesis. In this study, we have considerably expanded the phylogenetic diversity of known rumen bacteriocins and related peptides, identifying 4,326 putative bacteriocins, sactipeptide, lanthipeptide, and lassopeptide clusters. We examined the diversity of rumen lanthipeptides relative to lanthipeptides from cultured isolates. This analysis revealed several novel class II lanthipeptides encoded by rumen MAGs from diverse taxa that were predominantly found in *Eubacterium* and *Streptococcus*. Class II lanthipeptides contain a multifunctional lanthipeptide synthetase (LanM) that carries out both the dehydration and cyclization reactions[31]. LanM can possess high substrate tolerance[46,47] leading to diverse products and are currently used as promising targets for bioengineering[48]. Future research focusing on detailing the genetic organization and precursor peptide diversity of rumen class II lanthipeptides is needed as the rumen may be an untapped resource for novel class II lanthipeptides. The fraction of Hungate Collection isolates coding for lanthipeptides relative to MAGs suggests these genomic regions may be relatively difficult to assemble or bin from metagenomic data. The high number of rumen isolates harboring lanthipeptides and the diversity of lanthipeptides in MAGs suggest the rumen may be a promising source for novel lanthipeptide discovery. Similarly, a recent analysis of lassopeptides from the Hungate isolate genomes suggests Firmicutes are capable of producing several novel lassopeptides[49]. The recovered MAGs increase the number of bacteriocins native to the rumen ecosystem and available for targeted isolation and functional screening to develop novel probiotics and alternatives for antibiotics in ruminant production.

Given the abundance of NRPS gene clusters harbored by recovered genomes, we explored the diversity of this family of natural products through a relational network based on the BiG-SCAPE implemented distance metric. The network analysis

confirmed that NRPS BGCs have immensely diverse gene content and highlighted that ~70% of the network edges were between BGCs of the same taxonomic family. We further identified 687 NRPS gene clusters encoded by 125 archaeal genomes. Archaeal NRPS have been described previously, notably in *Methanobrevibacter ruminantium*[50]. A 2014 genomic survey found only three instances of archaeal NRPS in classes Methanobacteria and Methanomicrobia[51], and a recent analysis identified 73 BGCs from 203 archaeal genomes[52]. Phylogenetic analyses suggest that archaeal NRPS were acquired through horizontal transfer from bacteria[50,51]. Our network analysis appears to support this hypothesis as we established that there are NRPS in Euryarchaeota with high similarity to NRPS in different Firmicutes families. Given the proposed roles of NRPS in signaling and intercellular communication in ecosystems, it has been suggested that methanogen NRPS may be involved in perpetrating syntrophic interactions that are important for inter-species hydrogen transfer[50]. We noted methanogen genomes of the same species often contain very similar NRPS gene clusters, while other closely related genomes could lack NRPS gene clusters altogether. As such, we hypothesize that methanogens without NRPS may typically exist as symbionts of protozoa or other microbes and have lost the need to produce the compound. It is difficult to confidently assess the expression of populations in low abundance, but future work should aim to establish the expression patterns of methanogen NRPS gene clusters under various conditions.

In addition to predicting thousands of BGCs from MAGs, we also demonstrated a subset of BGCs that were expressed in rumen samples from high and low efficient steers. The differentially expressed BGCs were mainly sactipeptides encoded by *Selenomonas* and aryl polyenes and NRPS encoded by *Prevotella*. Host-associated microbes may mediate important interactions through the production of secondary metabolites[53]. *Prevotella* and *Selenomonas* populations are often linked to feed efficiency. Our approach using genome-resolved metagenomics and organism-specific normalization suggests secondary metabolites may play a role in this association. Further, the findings fit the emerging hypothesis that inefficient cattle have higher microbial diversity and produce a broader range of less usable metabolites for the animal's energy needs[54,55].

Inter-species diversity of BGCs appeared to be high in the rumen, while sub-species microdiversity analyses suggest strain-level BGC diversity may be more constant across samples. The majority of genes within BGCs had similar nucleotide diversity as other genes, with a few outliers that displayed very high diversity. We know little regarding the relationship between genetic and functional diversity of BGCs in the rumen. As such, future work may focus on obtaining a better understanding of the evolutionary processes shaping the microdiversity patterns of BGCs. The mean genome-wide nucleotide diversity of sub-species MAGs was more different across breeds than it was for genes of BGCs, suggesting host genetics may influence microdiversity.

In this study, we have provided a phylogenomic characterization of rumen-specific genomes that may serve as a foundation for future in silico and laboratory experiments to better explore the rumen as a source for alternative peptides and metabolites to modulate rumen fermentation. The genomes reported here and in other recent genetic explorations of the rumen microbiome appear to only provide a glimpse into rumen microbial diversity. Moving forward, we anticipate using the combination of cultured and uncultured genomes to populate a bottom-up systems biology framework that advances towards mechanistic understandings and modeling dynamics of the rumen microbial ecosystem.

## Methods

**Rumen metagenomic datasets**. We used 412 metagenomes for assembly and metagenomic binning (Supplementary Table 1). Rumen metagenomic studies with sufficient depth and quality were identified from the Sequence Read Archive, European Nucleotide Archive, and MG-RAST in early 2018. All publicly available metagenomes were sequenced on Illumina next-generation sequencing platforms. The remaining metagenomic datasets were previously unpublished.

The first two unpublished metagenomic datasets were from an 84-day growing study utilizing 120 steers to compare the influence of forage quality on methane emissions and subsequent 125-day finishing study with 60 steers to investigate the impacts of dietary fat, monensin, nitrate, and sulfate supplementation on methane emissions at the University of Nebraska Agriculture Research and Development Center[56]. The University of Nebraska-Lincoln Institutional Animal Care and Use Committee approved animal care and management procedures. From the original 120 animals in the growing study, 23 animals across different treatment groups were randomly selected for metagenomic sequencing. Sixty of the steers were utilized in a finishing study to evaluate the influence of dietary nitrate and sulfate on methane emissions and animal performance. From this study, 27 animals across different treatment groups were selected randomly for metagenomic sequencing. In both studies, sampling was conducted via esophageal tubing and snap-frozen with liquid nitrogen. Total DNA was extracted from rumen samples with the PowerMax Soil DNA Isolation Kit (MO BIO Laboratories, Inc.) according to the manufacturer's protocols. Metagenomes were prepared with the Nextera XT DNA Library Prep Kit and sequenced on the Illumina HiSeq platform using 150 bp paired-end sequencing. Raw data from these two datasets is associated with NCBI BioProject PRJNA627299 (Supplementary Table 1).

Paz et al. characterized the rumen microbiomes of 125 heifers and 122 steers to identify bacterial operational taxonomic units linked to feed efficiency[57]. From this cohort, 16 steers displaying divergent feed efficiency phenotypes were selected for metagenomic sequencing. In brief, rumen samples were collected through esophageal tubing and snap-frozen in liquid nitrogen. Total DNA was extracted from rumen samples with the PowerMax Soil DNA Isolation Kit (MO BIO Laboratories, Inc.) according to the manufacturer's protocols. Metagenomes were prepared using the NEBNext Ultra II DNA Library Prep Kit (New England Biolabs) and sequenced on the Illumina MiSeq platform (600 cycles, MiSeq Reagent Kit v3). Raw data from this study is associated with NCBI BioProject PRJNA627251 (Supplementary Table 1).

**Quality control of metagenomes**. Initial quality control of sequencing reads and adapter trimming were performed using BBDuk of the BBTools software suite (version 38.16; parameters: ktrim = r, k = 23, mink = 11, hdist = 1)[58]. VSEARCH (version 2.0.3) was used to remove sequences based on the presence of ambiguous bases (-fastq_maxns 0), minimum read length (range from -fastq_minlen 36 to -fastq_minlen 100 based on sample median read length and sequencing technology), and the maximum expected error rate (-fastq_maxee_rate 0.02 or -fastq_maxee_rate 0.025 depending on quality of the sequencing data and technology)[59].

**Assembly and metagenomic binning**. Paired-end and single-end sequences from each sample were assembled independently with MEGAHIT (version 1.1.1; parameters: -min-contig-len 1000, -k-min 27, -k-step 10)[60]. No co-assemblies were performed. We applied a maximum k-mer size of 87 (-k-max 87) for samples in which the longest read length was ≤100 bp. For samples with longer read lengths, we employed a maximum k-mer size of 127 (-k-max 127). The single-sample assemblies were input for both single-sample and multi-sample binning strategies with MetaBAT followed by re-assembly and dereplication[61]. Reads from each sample were mapped to assembled contigs (minimap2, parameters: -ax sr)[62]. The resulting alignments were used to bin contigs with a minimum length of 2000 bp for single-sample binning and 2500 for multi-sample binning strategies. Due to the total size of the collected datasets, the multi-sample binning was conducted independently for cattle (312 metagenomes) and other ruminant metagenomic datasets (100 metagenomes). Estimates of the completeness and contamination of the resulting bins were assessed using the lineage-specific workflow (lineage_wf) of CheckM (version 1.0.11)[63]. Bins ≥50% complete were re-assembled with SPAdes (version 3.13.0; -careful parameter)[64]. MAGs stemming from the single-sample binning pipeline were re-assembled only with reads from that same sample. MAGs reconstructed through multi-sample binning were re-assembled from the sample with the most reads aligning to the bin and from all reads aligning to the bin. The quality of re-assembled bins was assessed with CheckM. The best assembly (original or re-assembly) was retained based on the dRep quality score, where Genome Quality = Completeness − (5 · Contamination) + (Contamination · (Strain Heterogeneity/100)) + 0.5 · (log (N50)[20]. Contigs with divergent genomic properties (GC content and tetranucleotide frequency) were identified and removed with RefineM to reduce genome bin contamination[23]. Refined genomes from single-sample and multi-sample binning strategies were pooled and dereplicated with dRep at a threshold of 99% ANI[20]. Genomes meeting the following thresholds were retained: dRep quality score ≥60; N50 ≥5 kbp; ≤500 contigs; genome size ≥500 kbp; CheckM contamination estimate ≤10%; and CheckM completeness estimate ≥75%. Near-complete genomes were defined as MAGs with CheckM completeness estimate ≥90%, CheckM contamination estimate ≤5%, and N50 ≥15 kbp.

**Taxonomic and functional annotations of MAGs**. Taxonomy was assigned to MAGs using the classify workflow (classify_wf) of the Genome Taxonomy Database Toolkit (GTDB-Tk 0.2.2) with the associated Genome Taxonomy Database (release 86 v3). In short, the GTDB-Tk classifies each genome based on ANI to a curated collection of reference genomes, placement in the bacterial or archaeal reference genome tree, and relative evolutionary distance. For consistency, genomes from the Hungate1000 project[3] and Stewart et al.[4,5] were also assigned taxonomy with the GTDB-Tk. Depending on the taxonomic annotation, MAGs were functionally annotated with Prokka by evoking either the -kingdom Bacteria or -kingdom Archaea parameter (version 1.13.7)[65]. Prokka annotations were used to sum the number and types of tRNAs and rRNAs in each MAG (Supplementary Data 1).

**Inference of genome trees**. Phylogenetic trees were inferred with near-complete genomes (CheckM completeness estimate ≥90%, CheckM contamination estimate ≤5%, and N50 ≥15 kbp) using the GTDB-Tk (default parameters for the identify, align, and infer commands). Anvi'o was used to visualize the resulting Newick trees and associated metadata (version 5.5)[66]. We estimated how well a MAG was represented in a sample by calculating the percent of a MAG's bases with at least 1× coverage in the sample. The mean number of bases in a MAG with at least 1× coverage is presented for each metagenomic study and was used to compute the hierarchical clustering of rumen metagenomic datasets (Euclidean distance and Ward linkage).

**Similarity of reconstructed MAGs to GTDB reference genomes, the Hungate1000 Collection, and rumen-specific MAGs**. Recent analyses support a 95% ANI threshold to delineate microbial species[67,68]. The ANI values of MAGs from the current study and genomes from the GTDB (a curated and dereplicated collection of 22,441 genomes in the GTDB-Tk FastANI database[22]), Hungate1000 project[3], and Stewart et al.[4,5] were compared in a pairwise fashion with FastANI (version 1.1)[67]. Genome pairs with ≥95% ANI were denoted as overlapping species between the datasets. We visualized the number of overlapping genomes between each pair of datasets with UpSetR[69,70]. Additionally, genomes from the current study, the Hungate1000 project[3], and Stewart et al.[4,5] were clustered at 95% ANI thresholds with dRep[20] to approximate the number of microbial species represented across the rumen genomic collections. The number of genomes belonging to each 95% ANI cluster was used to calculate rarefaction curves in which cluster counts were subsampled without replacement at steps of 500 genomes with 10 replications at each step (QIIME version 1.9)[71].

The average genome size of reconstructed MAGs was smaller than was observed in the Hungate1000 Collection. In order to provide a better comparison of genome sizes across similar species, we evaluated the adjusted genome sizes of MAGs and Hunagte1000 Collection genomes that belonged to the same 95% ANI cluster based on Pearson correlation and linear regression, where Adjusted Genome Size = Genome Size/(Completeness + Contamination).

**Classification of metagenomic reads**. Reads from the 412 rumen metagenomes used to assemble MAGs and reads from 16 samples of an independent cattle metagenomic dataset[26] not used in binning were classified with different databases to assess the value of the reconstructed MAGs to improve metagenomic read classification. Reads were classified with Kraken2 (version 2.0.7; default parameters)[25] using a combination of the Kraken2 standard database containing bacterial, archaeal, fungal, and protozoa RefSeq genomes, 410 genomes from the Hungate1000 project[3], 4,941 MAGs from Stewart et al.[4,5], and the 2,809 MAGs from the current study.

**Phylogenetic analysis of biosynthetic gene clusters**. BGCs were identified within MAGs, the Hungate1000 collection[3], and the Stewart et al. MAGs[4,5] using antiSMASH 4.0)[27]. A network was constructed based on the BiG-SCAPE calculated distances between two BGCs (version "20190604")[29]. In short, BiG-SCAPE combines three approaches to measure the similarity of BGC pairs: (1) the Jaccard Index, which measures the percentage of shared domain types; (2) the Domain Sequence Similarity index that takes into account differences in Pfam domain copy number and sequence identity; (3) the Adjacency Index, a measure of the pairs of adjacent domains that are shared between BGCs. The raw BiG-SCAPE distances were converted to similarities for all analyses. Only NRPS ≥10 kbp (71.6% were ≥10 kbp) were evaluated and the network analysis was limited to Bacteroidota, Firmicutes, and Euryarchaeota phyla because these three phyla coded for 96.4% of NRPS gene clusters. Two BGCs (nodes in the network) were connected with an edge if the pairwise similarity was ≥0.3. We visualized the network as a hive plot with the R tidygraph package to demonstrate the inter- and intra-phylum diversity of NRPS BGCs. Nodes on an axis were ordered by the family of the genome coding the NRPS. Archaeal NRPS were further evaluated by placing BGCs into clusters based on a BiG-SCAPE glocal similarity threshold of 0.75. The distance between clusters was calculated as the mean pairwise similarity between the BGCs of two clusters. The resulting distance matrix was clustered with hierarchical clustering to produce a Newick tree (Euclidean distance and Ward linkage). The number of NRPS from near-complete archaeal genomes (CheckM completeness estimate ≥90%, CheckM contamination estimate ≤5%, and N50 ≥15 kbp) that belong to

each BiG-SCAPE cluster were tabulated and visualized alongside a phylogenetic tree inferred with the GTDB-Tk (default parameters for the identify, align, and infer commands)[22,63]. The data were visualized with Anvi'o (version 5.5)[66].

Lanthipeptide BGCs from the Hungate1000 genomes, Stewart et al. MAGs, and MAGs from the current study were clustered with lanthipeptide BGCs from RefSeq complete and draft isolate genomes. As part of BiG-FAM (version 1.0)[36], the RiPP BGCs of RefSeq genomes were made available at: https://doi.org/10.5281/zenodo.4106680[37] (predicted with antiSMASH verison 5.0). BiG-SCAPE (version "20190604"; –mode auto –cutoffs 0.3)[29] was used to cluster rumen and RefSeq lanthipeptide BGCs into GCFs and construct the relational network. Lanthipeptide classes were predicted with antiSMASH (version 5.1.2)[27]. A phylogenetic tree of near-complete rumen bacterial genomes encoding lanthipeptides was inferred with the GTDB-Tk (default parameters for the identify, align, and infer commands)[22,63]. The tree and associated data were visualized with Anvi'o (version 5.5)[66].

Rumen metatranscriptomic data[72] sequenced from steers with high (10 samples) and low (10 samples) residual feed intake were used to assess the expression of rumen microbial BGCs. ORF abundances for all rumen genomes were quantified with kallisto (version 0.45.0; default parameters)[73]. Kallisto generates pseudo-alignments based on exact k-mer matches. Differences in expression may be attributed to both variations in organism abundance and changes in microbial behavior under different conditions. Taxon-specific scaling of count data should reduce the influence of taxonomic composition changes[38]. Thus, to account for variations in taxonomic composition, count data for each genome were first partitioned and normalized separately with DESeq2 (version 1.24.0)[39]. The genome-specific normalization factors were used to scale raw BGC abundances from the same genome. Normalized BGC counts from each genome were re-combined to identify differentially expressed clusters between steers with high and low feed efficiency with DESeq2[39]. Only genomes with at least one read in all 20 samples (6,630 genomes) and BGCs with a minimum count of 100 reads were included in the analysis (648 BGCs).

Microdiversity analyses were carried out with InStrain (version 1.2.4)[40]. Reads from the 282 Illumina metagenomes described in Stewart et al. were mapped to the 4,941 MAGs previously recovered[4,5]. MAGs with an unmasked breadth ≥0.5 (i.e., ≥50% of the genome has 5× coverage) in a sample were considered to be present in that sample. That is, only genes from detected MAGs were used in subsequent analyses. Of the 4,941 MAGs, 2,926 had an unmasked breadth ≥0.5 in at least one sample. Further, genes were only considered present if they had ≥5× coverage in a sample in which the MAG was detected. The profile module of InStrain calculates the nucleotide diversity of scaffolds within a given sample. InStrain can use this profile to calculate the nucleotide diversity of genes (profile_genes module) and the mean nucleotide diversity of the genome (genome_wide module). The nucleotide diversity of detected genes was normalized based on the mean genome-wide microdiversity of the MAG (gene nucleotide diversity/genome-wide microdiversity) to reduce lineage-specific effects when comparing the microdiversity of BGCs. The normalized gene nucleotide diversity represents the nucleotide diversity of the gene relative to the nucleotide diversity of the rest of the genome. Statistical differences were assessed with Kruskal–Wallis and Wilcoxon rank-sum tests. All statistical comparisons were also carried out using raw nucleotide diversity values.

**Statistics and reproducibility**. We aligned rumen metatranscriptome data from steers characterized as high and low feed efficiency to the BGCs[72]. Read count data were normalized independently for each genome to better account for the variation in taxonomic composition across samples, as demonstrated previously[38]. Only genomes with at least one read in all 20 samples (6,630 genomes) and BGCs with a minimum count of 100 reads were included in the analysis (648 BGCs). Differentially expressed gene clusters between steers with high and low feed efficiency were identified with DESeq2[39] and false discovery rate adjusted $P < 0.05$ was considered statistically significant.

We also compared the raw and normalized nucleotide diversity (defined above) between the following groups: BGCs of different BGC classes, BGCs and other genes, and BGCs in four different breeds of cattle (Aberdeen Angus, Limousin, Charolais and Luing). The reads and MAGs for this analysis were previously described in the Stewart et al. studies[4,5]. Reads from the 282 metagenomes were mapped to the 4,941 MAGs and used as input to InStrain[40] to calculate the nucleotide diversity of a MAG or gene in a given sample. Statistical differences between groups were assessed with Kruskal–Wallis and Wilcoxon rank-sum tests in R and $P < 0.05$ was considered statistically significant.

**Reporting summary**. Further information on research design is available in the Nature Research Reporting Summary linked to this article.

## Data availability
The accessions for all metagenomes analyzed are available in Supplementary Table 1. Metagenomes previously not publicly available were deposited under NCBI BioProject PRJNA627299 and PRJNA627251. The 2809 reconstructed MAGs are available at: https://doi.org/10.6084/m9.figshare.12164250[74]. The authors declare that all other data supporting the findings of this study are available in the supplementary data files.

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

## Acknowledgements
This work was completed using the Holland Computing Center of the University of Nebraska, which receives support from the Nebraska Research Initiative. This study was partially supported by USDA National Institute of Food and Agriculture Fellowship 2016-67011-24783 (C.L.A.). This work was also supported by Animal Nutrition, Growth and Lactation grant no. 2018-67015-27496, Effective Mitigation Strategies for Anti-microbial Resistance grant no. 2018-68003-27545, and Multi-state research project accession no. 1000579 from the USDA National Institute of Food and Agriculture awarded to S.C.F.

## Author contributions
C.L.A. designed the study, conducted the analyses, and wrote the manuscript. S.C.F. designed the study, interpreted results, and wrote the manuscript.

## Competing interests
The authors declare the following competing interests: S.C.F. has disclosed a significant stake in NuGUT LLC. In accordance with its Conflict of Interest policy, the University of Nebraska-Lincoln's Conflict of Interest in Research Committee has determined that this must be disclosed. The remaining authors declare no competing interests.
