## [Peer Review File · Communications Biology]

Reviewers' Comments:

Reviewer #1:

Remarks to the Author:

The authors present new MAGs from various rumen datasets, compare them to existing genomic datasets, and go on to analyse biosynthetic gene clusters

I would ask for clarity first of all that these data sets have been included in the comparison of MAGs:

- MAGs from Solden, L. M. et al. Interspecies cross-feeding orchestrates carbon degradation in the rumen ecosystem. *Nat. Microbiol.* 3, 1274–1284 (2018)
- MAGs from Svartström, O. et al. Ninety-nine de novo assembled genomes from the moose (*Alces alces*) rumen microbiome provide new insights into microbial plant biomass degradation. *ISME J.* 11, 2538–2551 (2017).
- rumen MAGs from Parks, D. H. et al. Recovery of nearly 8,000 metagenome-assembled genomes substantially expands the tree of life. *Nat. Microbiol.* 2, 1533–1542 (2017), which used data from Wallace RJ, Rooke JA, McKain N, et al. The rumen microbial metagenome associated with high methane production in cattle. *BMC Genomics.* 2015;16:839.
- MAGs from Glendinning et al Metagenomic analysis of the cow, sheep, reindeer and red deer rumen, BioRxiv

The latter is particularly important as it includes deer, sheep and reindeer MAGs

Overall I think the authors could make it clearer exactly how many new MAGs and species this paper makes available, especially as they include data from Stewart et al 2018 in their datasets.

Line 29 - suggest replacing this with a less controversial statement e.g. "the majority of rumen microbial species remain uncultured". Given successful cuturomics studies in other species, I doubt the rumen is "special" in terms of its culturability.

Line 71 - I think the MIMAG criteria also includes tRNA genes?

Line 350 - no co-assemblies were performed. Stewart et al 2018 presented convincing evidence that co-assemblies resulted in the assembly of low abundance genomes. Why were no co-assemblies performed?

Line 356 - different cut-offs for contigs - 2000bp and 2500bp - why?

Line 373 - authors use a 75% completeness cut-off, yet Stewart et al used 80%. Should the authors not use the same cut-offs prior to comparison?

Line 381 - the GTBD classification of Hungate and Stewart collections would be useful, these should be included as supplements

Reviewer #2:

Remarks to the Author:

This is a well written and constructed paper containing information of interest for researchers interested in rumen-associated microbiomes and complex microbial ecosystems in general. In this work metagenome-assembled genomes (MAGs) from the rumen were constructed using publicly available metagenomes from various ruminant species as well as unpublished datasets from cattle

produced by the authors. Two thousand eight hundred and nine MAGs were assembled, expanding the catalogue of rumen MAGs as up to one third are novel. However, the most original contribution of the work is the identification of biosynthetic gene clusters present in rumen prokaryote genomes and I would suggest that the manuscript highlights preferentially this part of the work.

For improving the impact of the work, the authors should follow the recommendations for 'minimal information about a biosynthetic gene cluster' <https://dx.doi.org/10.1038/nchembio.1890> or justify why the guideline was not used.

Minor comments

1- L48 antimicrobial feed additives are 1) not allowed in many regions of the world and 2) not physiologically necessary for the holobiont (host and/or microbiota) to function. Please modify

2- Fig 5 c: small and not easy to understand; colour code seems not to be the same as Fig 5 a; nodes names?

3- Some citations might be missed: <https://doi.org/10.1093/femsec/fiz198> Check for other oversights

D. Morgavi

We appreciate the thoughtful feedback from the reviewers. We believe incorporating their feedback has resulted in an improved manuscript, especially in terms of clarity to the reader. Further, we have emphasized that the findings on biosynthetic gene cluster diversity are the central elements of the manuscript and have incorporated a new analysis of rumen lanthipeptide diversity. Below, the reviewer feedback is in bold followed by our response.

Reviewers' comments:

Reviewer #1 (Remarks to the Author):

The authors present new MAGs from various rumen datasets, compare them to existing genomic datasets, and go on to analyse biosynthetic gene clusters

I would ask for clarity first of all that these data sets have been included in the comparison of MAGs:

- **MAGs from Solden, L. M. et al. Interspecies cross-feeding orchestrates carbon degradation in the rumen ecosystem. Nat. Microbiol. 3, 1274–1284 (2018)**
- **MAGs from Svartström, O. et al. Ninety-nine de novo assembled genomes from the moose (*Alces alces*) rumen microbiome provide new insights into microbial plant biomass degradation. ISME J. 11, 2538–2551 (2017).**
- **rumen MAGs from Parks, D. H. et al. Recovery of nearly 8,000 metagenome-assembled genomes substantially expands the tree of life. Nat. Microbiol. 2, 1533–1542 (2017), which used data from Wallace RJ, Rooke JA, McKain N, et al. The rumen microbial metagenome associated with high methane production in cattle. BMC Genomics. 2015;16:839.**
- **MAGs from Glendinning et al Metagenomic analysis of the cow, sheep, reindeer and red deer rumen, BioRxiv**

The latter is particularly important as it includes deer, sheep and reindeer MAGs

The raw data from of the studies mentioned above were included in our binning efforts in the current manuscript (see Supplementary Table 1).

However, we did not compare the MAGs generated in this study with MAGs reported in the studies described above as it is likely that we will observe high overlap. However, the different assembly and binning approaches we have implemented would have likely resulted in more and improved bins as we binned across numerous other rumen metagenomic studies. In particular, there is evidence that binning contigs across a larger number of samples results in improved MAG quality (see comparisons in Pasoli *et al.* 2019 [10.1016/j.cell.2019.01.001](https://doi.org/10.1016/j.cell.2019.01.001)). Further, while the above papers will contain some of the MAGs that we report, some of the studies fail to make the MAGs available (Solden *et al.* and Svartström *et al.*, for instance make the raw data and assemblies available but not the MAGs).

Additionally, our aim in this manuscript was not to provide a comprehensive comparison of the recovered MAGs to all other rumen MAGs previously described but to leverage previous datasets in combination with new data to describe the phylogenetic diversity of biosynthetic gene clusters found within rumen microbiomes. We used comparisons to the two largest rumen

genomic collections (Hungate1000 and MAGs presented in Stewart et al. papers), as well as the genomes in GTDB, as an indicator that 1) we recovered previously described rumen microbial diversity and what those microbial species were, and 2) that our MAG dataset includes rumen microbial species not described in these other large genomic collections, indicating the potential value of the MAGs (see lines 105-107). We have significantly expanded the manuscript section focused on the phylogenetic diversity of BGCs by including a detailed analysis of lanthipeptides and re-organized the results in an attempt to bring more focus to BGC diversity.

Overall I think the authors could make it clearer exactly how many new MAGs and species this paper makes available, especially as they include data from Stewart et al 2018 in their datasets.

It is difficult to ascribe exactly how many of our MAGs represent new species as it is still an open question how to best functionally define a species from genomic data (although the evidence for ~95% ANI thresholds seems to be accumulating). We described (line 99) that 1007, of the MAGs found in this study have less than 95% ANI to genomes in the Hungate collection, Stewart *et al.* MAGs, or genomes in GTDB (using previous support for 95% ANI as an approximate threshold for delineating species). The overlaps of the MAGs described in this manuscript with the three previously mentioned databases are also visualized in Figure 3a. Figure 3b depicts the total number of genomes per 95% ANI cluster, indicating that most clusters contain only a single genome. Together, these results suggest many more rumen microbial species are likely contained in these datasets, as is supported by Figure 3c, but cannot be completely captured through current metagenomic assembly and binning approaches. Additionally, we have provided the taxonomy for all 95% ANI clusters identified from genomes in the current manuscript, Hungate1000 collection, and Stewart et al. papers in Figures 3d and 3e, as well as Supplementary Table 3. In another analysis, we clustered all genomes from the Hungate1000 collection, Stewart et al. papers, and our MAGs at 95% ANI (nearly 9,000 rumen-specific genomes). This yielded, 3,541 95% ANI clusters (approximate species) from the large rumen genomic collections (line 110). Of these clusters, 2,024 contained a MAG from the current manuscript (line 111), providing an estimate for the total number of species captured (~two-thirds of all described rumen microbial species in these three large rumen microbial genome databases).

It is not well established at what percent ANI we could determine if a MAG is unique to our manuscript relative to the previously mentioned databases. Even if we have recovered the same MAGs, it is likely our binning approach using metagenomes from a greater number of samples and species would have resulted in improved MAG size and quality. While we could repeat the above analysis at a threshold of 99% ANI (rather than 95%), we do not feel that would add more value to the manuscript, and instead potentially make the results less clear to the reader.

Line 29 - suggest replacing this with a less controversial statement e.g. "the majority of rumen microbial species remain uncultured". Given successful cutuomics studies in other species, I doubt the rumen is "special" in terms of its culturability.

We appreciate the reviewer bringing this to our attention if the wording was not clear – our intention was not to suggest that these microbes cannot be cultured, but rather than they remain

uncultured currently. We are of the opinion that any microbe can be brought into culture given the right conditions and that those conditions just have not been identified or attempted for the majority of species. We have changed the wording to make our intentions clearer (line 29-30).

Line 71 - I think the MIMAG criteria also includes tRNA genes?

Thanks for bringing this oversight to our attention. Yes, the MIMAG definition does include the number of tRNA genes present. The number of tRNA genes was considered when defining which MAGs meet the MIMAG standards and this information, along with other genomic properties, is included in Supplementary Table 2. We have updated the text to include that tRNA genes are an element of the MIMAG standard (lines 74-77).

Line 350 - no co-assemblies were performed. Stewart et al 2018 presented convincing evidence that co-assemblies resulted in the assembly of low abundance genomes. Why were no co-assemblies performed?

Thanks for bringing this important consideration to the forefront. Metagenomic co-assemblies are a trade-off between better assembling low abundant populations and introducing more strain-level variation that breaks assemblies. We believe the best evidence supporting that introducing more strain-level variation (which is the result of pooling samples for co-assembly) breaks assemblies comes from the CAMI paper (Sczyrba *et al.* 2017 doi: 10.1038/nmeth.4458) and Awad *et al.* (2017, <https://doi.org/10.1101/155358>). Both these papers demonstrated the concept of what they term “strain confusion” – as more strain variation is introduced into a collection of reads, there is often a significant loss in assembly quality and contiguity, even if the entire genome is present in the reads at sufficient coverage. Consider the following excerpt from Awad *et al.*:

“The Shewanella baltica OS185 genome is a good example: there are two strain variants, OS185 and OS223, present in the defined community. Both are present at more than 99% in the reads, and more than 98% in 51-mers, but only 75% of S. baltica OS185 and 50% of S. baltica OS223 are recovered by assemblers. This is a clear case of “strain confusion” where the assemblers simply fail to output contigs for a substantial portion of the two genomes.”

Our main reasoning for opting for single-sample assemblies was driven by these findings related to strain-variation breaking assemblies. However, we also viewed single-sample assemblies as more feasible for the scale of data used in this study. Performing a co-assembly of all of the rumen metagenomes, even for just cattle (335 metagenomes), was not computationally feasible. Additionally, we think there is good evidence that single-sample assemblies, especially in combination with re-assembly, perform better than co-assemblies on large-scale datasets, as outlined in Pasoli *et al.* (2019, [10.1016/j.cell.2019.01.001](https://doi.org/10.1016/j.cell.2019.01.001)). In this paper they compared the binning outputs of single-sample assembly and co-assembly. Through their rigorous comparisons, the authors reached the following conclusion regarding single-sample assembly: *“It is therefore more suitable for the very large scale analyses considered here where the aim is to generate a small number of HQ strains from each sample to provide the most comprehensive picture of overall diversity in the human gut.”*

And:

“Other genome quality statistics were very similar between the two approaches with however the co-assembly method showing slightly more contamination (1.7% against 0.9% for HQ genomes, Table S2). Overall, this suggests that large scale co-assembly may at best offer limited improvement in terms of overall recovered diversity.”

In summary, we opted for single-sample assemblies that reduce strain-level variation, but with a potential sacrifice of assembling some low abundant organisms. That being said, our binning and re-assembly strategy may have overcome this tradeoff. Low abundant organisms may be fragmented in one sample, but the fragmented contigs from multiple samples would be clustered into a single bin and then re-assembled into more contiguous assemblies through our mapping and re-assembly approach outlined in the methods. In fact, while this data was not displayed in the manuscript, the genome quality score of ~80% of MAGs were improved through re-assembly. Given the above evidence and lines of thinking, we believe single-sample assembly followed by single-sample and multi-sample binning, and re-assembly of resulting bins may result in slight decreases in completeness, but decreases contamination and reduces the influence of strain variation on assembly quality, while being scalable to the number of samples used in this study.

Line 356 - different cut-offs for contigs - 2000bp and 2500bp - why?

Different contig lengths, 2000 bp for single-sample binning and 2500 for multi-sample binning, were chosen for computational reasons. For cattle, we performed multi-sample binning of contigs across 335 samples, which is computationally demanding and was not feasible with our computational resources when using the 2000 bp cutoff. As these fragments are already relatively short, the difference in binning outputs between 2000 and 2500 is unlikely to have a meaningful influence on average MAG completeness across ~3000 MAGs and is further reduced by the re-assembly and dereplication strategy we employed. Both the 2000 and 2500 parameters are above the thresholds suggested for MetaBAT2 to bin MAGs out of complex metagenomes and we do not believe there is evidence suggesting that redoing the binning efforts with the same cutoffs will result in meaningful increases in completeness or reduction in contamination.

Line 373 - authors use a 75% completeness cut-off, yet Stewart et al used 80%. Should the authors not use the same cut-offs prior to comparison?

The intentions of our manuscript are not to be a rigorous comparison to the MAGs of Stewart *et al.*, or to other rumen genomic datasets, but rather as a starting place to highlight that while some overlap exists between the datasets, there are substantial differences in the composition of species recovered as well (see Figure 3). Further, Stewart *et al.* used an 80% completeness threshold, but our standards for contamination were much more rigorous, likely removing genomes that Stewart *et al.* would have included. Rather than simply using completeness and contamination thresholds, we used the Genome Quality metric proposed in Parks *et al.* (2017, [10.1038/s41564-017-0012-7](https://doi.org/10.1038/s41564-017-0012-7)):

Genome Quality = Completeness – (5 * Contamination) + (Contamination * (Strain * Heterogeneity / 100)) + 0.5 * (log(N50))

We then only considered MAGs meeting the following conditions:

dRep quality score ≥ 60 ; N50 ≥ 5 kbp; ≤ 500 contigs; genome size ≥ 500 kbp.

This is more rigorous than the completeness and contamination thresholds used in Stewart *et al.* Further, our approach places more weight on contamination, ensuring that contamination is low *relative* to the completeness of the MAG. This approach is also flexible, in that more complete MAGs would be allowed to have slightly higher contamination than less complete MAGs, rather than a simple 10% contamination threshold for all MAGs. For instance, if a MAG was 80% complete, likely the highest amount of contamination that would meet our inclusion criteria would be 4-5%. However, in the Stewart *et al.* work, MAGs with contamination rates up to 10% were included.

Overall, these thresholds are very arbitrary and just a means to define which MAGs are of sufficient quality for downstream analyses. We also feel that genomic collections are comparable even without having the exact same thresholds for quality. The goal of our manuscript was not to be a strict comparison to those MAGs in Stewart *et al.*, but to rather use those MAGs as a comparison to simply highlight the composition relative to some of the previous findings from the rumen before emphasizing the role of secondary metabolism in these MAGs. In the revised manuscript we have attempted to place more of an emphasis on the distribution of biosynthetic gene clusters and less on the comparisons to Stewart *et al* MAGs.

Line 381 - the GTDB classification of Hungate and Stewart collections would be useful, these should be included as supplements

Thanks, the reviewer is correct that this information would be helpful to others in the field and we did include it as part of Supplementary Table 3. This table includes the GTDB classifications for 8,160 rumen-specific genomes and MAGs and indicates which genomes belong to which 95% ANI clusters (approximate species threshold), as defined by dRep. Further, we provide the taxonomy of all rumen genomes and MAGs alongside their biosynthetic gene cluster information as part of Supplementary Table 4.

It should be noted that the GTDB classifications are always changing in light of new evidence and hopefully others will update this information in the future as changes to GTDB are made. A central repository with all rumen genomes and MAGs would be helpful to the field but would require a substantial time investment.

Reviewer #2 (Remarks to the Author):

This is a well written and constructed paper containing information of interest for researchers interested in rumen-associated microbiomes and complex microbial ecosystems in general. In this work metagenome-assembled genomes (MAGs) from the rumen were constructed using publicly available metagenomes from various ruminant species as well as unpublished datasets from cattle produced by the authors. Two thousand eight hundred and nine MAGs were assembled, expanding the catalogue of rumen MAGs as up to one third are novel. However, the most original contribution of the work is the identification of biosynthetic gene clusters present in rumen prokaryote genomes and I would suggest that the manuscript highlights preferentially this part of the work.

For improving the impact of the work, the authors should follow the recommendations for ‘minimal information about a biosynthetic gene cluster’ <https://dx.doi.org/10.1038/nchembio.1890> or justify why the guideline was not used.

To the reviewer’s point, to emphasize that the findings on BGC diversity are the central aspects of the manuscript, we have reorganized elements of the manuscript and added a detailed analysis of rumen lanthipeptide diversity (lines 183-215). The lanthipeptide analyses reveal the rumen MAGs encode many novel class II lanthipeptides not found in the Hungate isolates or other RefSeq genomes.

We appreciate the reviewer suggesting we apply the standards of the “minimal information about a biosynthetic gene cluster (MIBiG)” for the biosynthetic genes described in this manuscript – more data on MAGs should be encouraged to be cataloged in long-term repositories. The recent 2019 publication for MIBiG described updates to the database, including 851 BGCs added over the previous five years. This highlights that the database is highly curated for BGCs that have *known* chemical products and structures. While metagenomic assembly and binning has improved, they may still contain contamination and as a result, submitting BGCs derived from metagenomic binning to this database may not be appropriate. The value of MIBiG for metagenomes appears to be comparing assembled data to the curated MIBiG database to search for known BGCs. For example, Crits-Christoph et al. (2018, <https://doi.org/10.1038/s41586-018-0207-y>) and Bahram et al. (2018, <https://doi.org/10.1038/s41586-018-0386-6>) did exactly this, searching MAGs or metagenomic contigs from soil that matched clusters in MIBiG. This was included as part of our antiSMASH search; however, this was used to highlight the novelty of the putative BGCs encoded by rumen microbes as there were few hits to MIBiG. In neither of these publications did they submit the resulting metagenomic BGCs to MIBiG or report them in the MIBiG standard format. This highlights the standard of data required for MIBiG and that other high impact metagenomic publications on BGCs have not been added to the database, nor reported in the MIBiG style.

Additionally, the annotations we provide for BGCs meets the minimal information for MIBiG. Consider the entry below in MIBiG, which includes only the BGC class, compounds, and taxonomy information:

```

{
  "changelog": [
    {
      "comments": [
        "Submitted"
      ],
      "contributors": [
        "AAAAAAAAAAAAAAAAAAAAAAAA"
      ],
      "version": "1.0"
    },
    {
      "comments": [
        "Migrated from v1.4"
      ],
      "contributors": [
        "AAAAAAAAAAAAAAAAAAAAAAAA"
      ],
      "version": "2.0"
    }
  ],
  "cluster": {
    "biosyn_class": [
      "RiPP"
    ],
    "compounds": [
      {
        "compound": "bovicin HJ50"
      }
    ],
    "loci": {
      "accession": "EU497962.2",
      "completeness": "Unknown"
    },
    "mibig_accession": "BGC0000498",
    "minimal": true,
    "ncbi_tax_id": "1335",
    "organism_name": "Streptococcus equinus",
    "publications": [
      "pubmed:19202107"
    ],
    "ripp": {
      "subclass": "Lanthipeptide"
    }
  }
}

```

We believe that the information we include as part of Supplementary Table 4, goes beyond the minimum information required by MIBiG. As part of Supplementary Table 4, we include the following: BGC class, contig ID in the MAG, cluster number of the BGC in case the contig contains multiple BGCs, contig length, the ID of the ORFs contained in the BGCs, start and end positions of the BGC on the contig, and taxonomy of the MAG. This should provide adequate information for those that download the MAGs to easily extract the BGCs for further analysis.

Minor comments

1- L48 antimicrobial feed additives are 1) not allowed in many regions of the world and 2) not physiologically necessary for the holobiont (host and/or microbiota) to function. Please modify

Thanks for the feedback – we agree with this assessment. We did not intend for it to read that way. We have changed the wording to reflect that (lines 41-46). Our aim was to propose that research about the distribution of biosynthetic gene clusters might aid the development of alternatives to antibiotics for regions where antibiotics are used intensely, mainly for growth promotion and disease prevention.

2- Fig 5 c: small and not easy to understand; color code seems not to be the same as Fig5 a; nodes names?

Thanks for catching this – this is now Figure 4 in the new version of the manuscript. We have matched the colors in panel c to panel a, added the legend, and expanded the text description in

the figure caption in an effort to make this figure easier to understand. We feel the findings presented in this figure are of interest to understanding the evolution of methanogen metabolism and we appreciate the reviewer pushing us to make these results more interpretable.

3- Some citations might be missed: <https://doi.org/10.1093/femsec/fiz198> Check for other oversights

Thanks for the suggestion – this was a simple oversight as the manuscript was largely complete before this publication came out. The research is an excellent addition to the rumen microbiology field, and we appreciate the reviewer bringing it to our attention. We have added a statement about this work in the introduction (line 41).

Reviewers' Comments:

Reviewer #2:

Remarks to the Author:

The authors have satisfactorily addressed all comments.

No further major comments.

A minor detail for the added sentence in L48 of the revised manuscript "In support, a recent study of rumen metatranscriptomic data ..." A reference is missing.

Reviewer #3:

Remarks to the Author:

I am not sure I completely agree with the view that MAGs are not provided. The studies cited to back up this claim appear to have their MAGs publicly available (Solden et al), see (https://gold.jgi.doe.gov/analysis_projects?Study.GOLD%20Study%20ID=G0121650). Some of the related statements are also difficult to sustain. For instance, the authors make a statement that "In particular there is evidence that binning contigs across a larger number" For this to be supported, the available MAGs should be used to provide quantitative evidence and for comparisons to confirm whether their assembly and approach may result in improved bins as claimed. The authors also claim that their study was "not to provide a comprehensive comparison of the recovered MAGs....." but analysis such as clustering, ANI analysis etc raises questions. The observed high overlaps may be due to methodological differences. The authors cannot exclude the possibility that the methods used in the other studies may recover sets of MAGs which could be missed by their approach. The statement that "our MAG dataset includes rumen, microbial species not described in these other genomic collections" should be clarified. For instance, which collections are being referred to? In light of what I've shown earlier, I am not convinced that the 'genomic collections' are as extensive as claimed. A caveat is that sadly some people continue not to share their MAGs and it is understandable that the authors could not include these in their analysis.

The second MAG related comment by the reviewer is also not well addressed. In response to the question re the number of total new MAGs reported, the authors response is unclear. For instance, does the number include the removal of singletons? It would be interesting to know if MAGscan could be used for comparing other genomes such as those obtained through culture based methods or those from single cell genomics.

The reviewer also voiced concern regarding the lack of co-assemblies. We usually follow this approach for our data. I am not sure what the reluctance is on the part of the authors to use co-assemblies but prefer "multi-sample binning" and re-assembly of MAGs using reads (see lines 415-419) "This also seems to be in contrast to the comment made in the rebuttal that pooling samples results in "more strain variation"

The final comment in page 4 is also not sufficiently convincing and speculative. The authors do not provide a response to the question re the cut off for contigs. The response that they 'do not believe' there is evidence that redoing the bins with the same cut-offs may increase the quality of the data is not sufficient. At the very least, the authors could demonstrate this with a smaller dataset and validly show that there is no increase.

Overall, I am not sufficiently convinced by the rebuttal.

Reviewers' comments:

Reviewer #2 (Remarks to the Author):

The authors have satisfactorily addressed all comments.

No further major comments.

A minor detail for the added sentence in L48 of the revised manuscript "In support, a recent study of rumen metatranscriptomic data ..." A reference is missing

Thanks for noting this - the reference has been added to the manuscript.

Reviewer #3 (Remarks to the Author):

I am not sure I completely agree with the view that MAGs are not provided. The studies cited to back up this claim appear to have their MAGs publicly available (Solden et al), see https://gold.jgi.doe.gov/analysis_projects?Study.GOLD%20Study%20ID=G0121650). Some of the related statements are also difficult to sustain. For instance, the authors make a statement that "In particular there is evidence that binning contigs across a larger number". For this to be supported, the available MAGs should be used to provide quantitative evidence and for comparisons to confirm whether their assembly and approach may result in improved bins as claimed. The authors also claim that their study was "not to provide a comprehensive comparison of the recovered MAGs....." but analysis such as clustering, ANI analysis etc raises questions. The observed high overlaps may be due to methodological differences. The authors cannot exclude the possibility that the methods used in the other studies may recover sets of MAGs which could be missed by their approach. The statement that "our MAG dataset includes rumen, microbial species not described in these other genomic collections" should be clarified. For instance, which collections are being referred to? In light of what I've shown earlier, I am not convinced that the 'genomic collections' are as extensive as claimed. A caveat is that sadly some people continue not to share their MAGs and it is understandable that the authors could not include these in their analysis.

We appreciate pointing out that the 77 MAGs from the Solden et al. paper are available via JGI. The 2018 Solden et al manuscript containing the MAGs was a follow-up to the initial 2016 paper which contained the same metagenomic data and no MAGs -

<https://www.nature.com/articles/ismej2016150>. This was an oversight as we had written a draft of the manuscript prior to the release of the 2018 Solden et al manuscript. However, there is a relatively small number of MAGs in this paper and the other papers whose public metagenomic data were used in the current manuscript (see table below) relative to the Stewart et al dataset (~5,000 MAGs) and the MAGs in our current manuscript (~3,000 MAGs). While the reviewer disagrees, we stand by that the purpose of the current manuscript was to leverage existing

rumen metagenomic datasets to construct reference genomes for rumen species lacking genomic representation and link these genomes to potential secondary metabolite gene clusters, and highlight how these MAGs improve our understanding of rumen genomic microdiversity and feed efficiency in cattle. Comparing the MAGs from the current study to every other rumen MAG paper would be outside the scope of this work and we instead focused on comparing the MAGs in the current manuscript to the Stewart et al datasets (~5,000 MAGs), the Hungate rumen isolates (410 genomes) and GTDB species-level reference genomes (~22,000 genomes) to provide the reader with an idea of how novel the MAGs in our dataset were to the largest rumen genomic collections - Hungate isolate genomes and Stewart et al MAGS - and to reference isolate genomes (GTDB includes GenBank and RefSeq genomes). The results of these comparisons are shown in Figure 3 of the manuscript. However, we do agree that we cannot exclude the possibility that previous papers may have yielded MAGs missed by our approach and we adjusted a sentence in the manuscript to reflect this more clearly (line 276-278). Additionally, we decided to change the wording in the manuscript claiming 1,007 species are novel to this manuscript (more detail on this number in another response below), to reflect that we can only say these species do not exist in the Stewart et al datasets, Hungate rumen isolate collection, or in GTDB, which together do encompass a very large portion of known rumen microbial diversity, but perhaps not completely (lines 100-101). We believe the text is clear about which databases and how many genomes we compared the MAGs to (line 94-96), *“We clustered genomes based on approximate species-level thresholds ($\geq 95\%$ ANI) and calculated the intersection between MAGs in the current study and the Hungate1000 Collection (410 genomes), MAGs from Stewart et al. (4,941 genomes) and a dereplicated genome collection from the GTDB (22,441 genomes, see methods).”*

Ruminant	QC Bases (Gbp)	Sam ples	Databa se	Accession	Publication DOI	Number of MAGs reported
Bison	52.3	8	NCBI BioProject	PRJNA214227	NA	No MAGs
Cattle	22.8	16	NCBI BioProject	PRJNA627251	Current Study	No MAGs
Cattle	43.5	27	NCBI BioProject	PRJNA627299	Current Study	No MAGs
Cattle	33.3	23	NCBI BioProject	PRJNA627299	Current Study	No MAGs
Cattle	166.0	1	NCBI	PRJNA60251	https://doi.org/1	15 MAGs - are available.

			BioProject		0.1126/science.1200387	Also present in GTDB.
Cattle	85.1	8	NCBI BioProject	PRJEB10338	https://doi.org/10.1186/s12864-015-2032-0	No MAGs
Cattle	52.7	64	MG-RAST	mgp4126	https://doi.org/10.1186/1471-2156-13-53	No MAGs
Cattle	12.5	14	NCBI BioProject	PRJEB8939	https://doi.org/10.1038/ismej.2016.172	No MAGs
Cattle	60.6	9	NCBI BioProject	PRJNA322715	https://doi.org/10.1007/s00203-016-1311-8	No MAGs
Cattle	4.8	1	NCBI BioProject	PRJNA270714	https://doi.org/10.1128/genomeA.00723-15	No MAGs
Cattle	20.9	2	NCBI BioProject	PRJNA291523	https://doi.org/10.1093/nar/gkv973	No MAGs
Cattle	125.0	16	NCBI BioProject	PRJNA214227	NA	No MAGs
Cattle	666.0	42	NCBI BioProject	PRJEB21624	https://doi.org/10.1038/s41467-018-03317-6	913 MAGs - Stewart et al first paper
Cattle	73.0	7	NCBI BioProject	PRJNA319009	https://doi.org/10.1016/j.scitotenv.2017.01.096	No MAGs
Cattle	46.7	9	NCBI BioProject	PRJNA322715	https://doi.org/10.1007/s00203-016-1311-8	No MAGs
Cattle	12.4	14	NCBI BioProject	PRJEB8939	https://doi.org/10.1038/ismej.2016.172	No MAGs
Cattle	762.7	82	NCBI BioProject	PRJEB23561	https://doi.org/10.1101/272690	324 MAGs; 196 >50% complete - Made

			ect			available May 2020
Deer (White-tailed)	34.2	4	NCBI BioProj ect	PRJNA214227	NA	No MAGs
Deer (Red)	28.6	4	NCBI BioProj ect	PRJNA214227	NA	No MAGs
Moose	40.9	3	NCBI BioProj ect	PRJNA301235	https://doi.org/10.1038/ismej.2016.150	77 MAGs
Moose	67.9	6	NCBI BioProj ect	PRJEB12797	https://doi.org/10.1038/ismej.2017.108	99 MAGs
Sheep	113.5	16	NCBI BioProj ect	PRJNA214227	https://doi.org/10.1093/dnares/dst044	No MAGs
Sheep	118.3	39	MG- RAST	mgp7948, mgp7949, mgp7950, mgp7957, mgp7958, mgp7959, mgp7960, mgp7961, mgp7962, mgp7963, mgp7964, mgp7965, mgp7966, mgp7967, mgp7968, mgp7969, mgp7970, mgp7974, mgp7975, mgp8090, mgp8091, mgp8092,	https://doi.org/10.1371/journal.pone.0110505	No MAGs

				mgp8093, mgp8094, mgp8095, mgp8096, mgp8097, mgp8098, mgp8099, mgp8108, mgp8109, mgp8110, mgp8111, mgp8112, mgp8113, mgp8114, mgp8115, mgp8116, mgp8117		
Sheep	656.5	20	NCBI BioProj ect	PRJNA202380	https://doi.org/10.1101/gr.168245.113	No MAGs

Many of the reviewer's comments here and below center around asking us to perform two analyses:

- 1) A quantitative analysis that demonstrates single-sample assembly followed by multi-sample binning and MAG reassembly results in improved bins in comparison to co-assembly followed by binning
- 2) An analysis demonstrating that a 2500 bp contig length threshold is suitable for multi-sample binning because we used 2000 bp contig length threshold for single-sample binning

However, in both of these situations, we did not choose the alternative because it was not computationally feasible on our university cluster due to the scale of the data, i.e. the analysis could not be completed with 1TB RAM and 64 threads within 168 hours of compute time - a relatively large resource. We appreciate the reviewer asking for experiments to support our bioinformatic decisions, but we find these additional experiments to be an unnecessary ask and would not change or strengthen the conclusions of the manuscript, as even if an analysis did demonstrate that co-assembly followed by binning is better for our specific dataset, negating the

same comparisons already done on human metagenomes (see below), we could not implement it.

Below, we describe in more detail our response to the first point. We have provided a response to the second point in reply to the final critique raised by the reviewer.

The literature supports that both co-assembly and single-sample assembly followed by multi-sample binning each have advantages and disadvantages. Co-assembly can recover more rare species; however, this is only true if that given species does not have extensive strain variation, leading to fragmented assemblies. If you pool multiple samples prior to co-assembly, this generates more strain diversity (more closely related genomes), which in turn breaks assemblies rather than improving them for species with high intra-species diversity, as demonstrated clearly in Awad et al (<https://doi.org/10.1101/155358>). By assembling single samples and in turn binning contigs across all samples (followed by read mapping and reassembly of MAGs), you avoid the effects of increased strain diversity caused by the pooling of samples.

We were not reluctant to do co-assembly, but rather it was not computationally feasible to co-assemble 3.3 Tbp of data across 435 metagenomes (MEGAHIT, 1TB RAM, 64 threads, 1 week compute time). When co-assemblies are not feasible, it was previously thought to be best to subsample or normalize the data (i.e. digital normalization - cite). Instead, Pasolli et al (<https://doi.org/10.1016/j.cell.2019.01.001>) demonstrated that single-sample assembly of human gut metagenomes (154,723 MAGs from 9,428 metagenomes) followed by binning contigs across all samples produces, on average, bins of the same quality as co-assembly followed by binning, while remaining computationally feasible for large datasets. A similar approach was used by Nayfach et al in the assembly of ~60,000 MAGs from 3,810 human gut metagenomes (<https://doi.org/10.1038/s41586-019-1058-x>). Repeating the analysis of Pasolli et al would not strengthen the conclusions of the current manuscript and is not a worthwhile investment of time, as there is no reason to believe these findings from human gut metagenomes would not extend to rumen metagenomes. In summary, the literature supports that co-assembly and single-sample assembly both have merits under different circumstances. Ideally, we could have merged the results of both strategies, a solution more researchers should employ. However, it was not computationally feasible and therefore we adopted our strategy of single-sample assembly followed by multi-sample binning, which is supported by the comparisons and previous findings in Pasolli et al to generate MAGs of similar quality to co-assembly binning.

The second MAG related comment by the reviewer is also not well addressed. In response to the question re the number of total new MAGs reported, the authors response is unclear. For instance, does the number include the removal of singletons? It would be interesting to know if MAGscan could be used for comparing other genomes such as those obtained through culture based methods or those from single cell genomics.

Is MAGscan a computational tool? We don't see a reference to it in the literature. We responded to the above critique assuming that MAGscan was a typo and that the sentence should read .. "to know if MAGs can be used..."

To clarify we believe the reviewer is referring to singleton MAGs as those MAGs that were the only genomic representation of a species (95% ANI cluster). Approximately 60% of the MAGs are the single genomic reference for species (95% ANI cluster). None of these MAGs have been removed from the analysis and as these MAGs are clearly real population genomes, as supported by checkM and other analyses that demonstrate the MAGs exist in multiple rumen metagenomes (i.e. coverage breadth with InStrain, Kraken, etc). Instead, this and the rarefaction curve (Fig. 3C) suggests the 8,160 rumen genomes we analyzed in combination to our MAGs only represent a fraction of the estimated microbial species diversity in the rumen. We go on to demonstrate that 1,007 MAGs from our study are representative of species not present in GTDB (includes Refseq, GenBank, some environmental MAGs) the Stewart et al rumen MAGs, or the Hungate rumen isolate collection, as these MAGs do not share >95% ANI with a genome or MAG in these genomic databases / collections. There are no established ANI thresholds to assess how many unique strains our MAGs represent.

Yes, MAGs in general have been extensively compared to genomes obtained through typical culturing and through SAGs. This is how GTDB assigns taxonomy to MAGs - by placing the MAG within the phylogenetic context of reference genomes, which are almost exclusively isolate genomes. The MAGs vs SAGs comparison is in particular interesting as MAGs are a population genome and SAGs are by definition not - see: <https://microbiomejournal.biomedcentral.com/articles/10.1186/s40168-018-0550-0> and <https://peerj.com/articles/10119/>. As a result, that is also what we have done here by comparing our MAGs to the GTDB database, which includes rumen isolates that are in GenBank and RefSeq, the Hungate rumen isolates, and the Stewart et al MAGs.

The reviewer also voiced concern regarding the lack of co-assemblies. We usually follow this approach for our data. I am not sure what the reluctance is on the part of the authors to use co-assemblies but prefer "multi-sample binning" and re-assembly of MAGs using reads (see lines 415-419) "This also seems to be in contrast to the comment made in the rebuttal that pooling samples results in "more strain variation"

Addressed in the first response to this reviewer.

The final comment in page 4 is also not sufficiently convincing and speculative. The authors do not provide a response to the question re the cut off for contigs. The response that they 'do not believe' there is evidence that redoing the bins with the same cut-offs may increase the quality of the data is not sufficient. At the very least, the authors could demonstrate this with a smaller dataset and validly show that there is no increase.

The original question stated by the reviewer is why was a 2000 bp cutoff used when binning contigs within single samples and a 2500 bp cutoff used when binning those single sample assembly contigs across multiple assemblies - "Line 356 - different cut-offs for contigs - 2000bp and 2500bp - why?"

The reason was that it was not computationally feasible to bin that contigs when using a 2000 bp threshold across 335 cattle metagenomes (recall, multi-sample binning was conducted across two sets of samples, cattle only and all other ruminants), but it was feasible when reducing the number of contigs by employing a 2500 bp length threshold. From the literature, it is well understood that using short contigs results in more MAG contamination, but there are no established thresholds, as the results may vary from one dataset to the next. From the Metabat paper - cite: "...suggest that the values of two parameters of the model, b and c are unstable if the size of either contig is very small (<2 kb) and one should be cautious to allow smaller contigs to be binned." Instead, the common practice is to refine MAGs (which we did in this manuscript) and remove MAGs not meeting completion and contamination thresholds.

Nonetheless, we have some data comparing multi-sample binning using contigs >2000 and >2500 bp from the 100 non-cattle ruminant metagenomes in the current study. After binning, we summarized the number of MAGs with >70% completion and <10% contamination - no MAG refining or reassembly were applied, which was done in the manuscript to improve MAG quality. The 2000 bp contig threshold resulted in 1313 MAGs while the 2500 bp threshold produced 1199 MAGs. The median contamination of the MAGs with contigs >2500 bp was ~2% higher. While the 2000 bp contig threshold results in slightly more MAGs, this does not invalidate using 2500 bp contigs. Further, the results of the two approaches, single-sample and multi-sample binning, were later merged and dereplicated. Overall, it was not possible to bin all contigs >2000 bp due to the size of the 335 cattle metagenomic samples, and using 2500 bp contig vs 2000 bp contig threshold has very little bearing on the overall quality of the MAGs.

REVIEWERS' COMMENTS:

Reviewer #3 (Remarks to the Author):

The authors have provided a comprehensive and detailed response to some of my serious concerns. Although I am unhappy with aspects of the rebuttal (detailed below) I believe the authors have attempted to address my concerns within their computational constraints.

The first major issue, regarding the availability of MAGs from the Solden et al paper is understandable, given the fact that their ms was prepared prior to the release of the paper. I also appreciate the rewording to reflect that the authors cannot claim novelty outside of the reported dataset by Stewart et al.

Co-assemblies are generally accepted across the community. Although the response to this query is verbose it is not sufficiently convincing. Although I appreciate the fact that some analysis could not be done using the servers at their institution, I am not sure this in itself is a compelling.

The authors are correct that my second comment contained a typo. I was indeed interested in knowing whether MAGs can be used for valid comparisons with other genomes. Thanks for the response, and the links to previous literature on this.

REVIEWERS' COMMENTS:

Reviewer #3 (Remarks to the Author):

The authors have provided a comprehensive and detailed response to some of my serious concerns. Although I am unhappy with aspects of the rebuttal (detailed below) I believe the authors have attempted to address my concerns within their computational constraints.

The first major issue, regarding the availability of MAGs from the Solden et al paper is understandable, given the fact that their ms was prepared prior to the release of the paper. I also appreciate the rewording to reflect that the authors cannot claim novelty outside of the reported dataset by Stewart et al.

Thanks for the response and well thought out suggestions.

Co-assemblies are generally accepted across the community. Although the response to this query is verbose it is not sufficiently convincing. Although I appreciate the fact that some analysis could not be done using the servers at their institution, I am not sure this in itself is a compelling.

We agree that metagenomic co-assemblies can provide benefits, such as improved assemblies of low abundant organisms. However, very large metagenomic datasets, such as in this study, make it computationally challenging if not impossible to do co-assemblies. As a result, single-sample assemblies have been employed, as was done in the current manuscript. In particular, Nayfach et al. (<https://doi.org/10.1038/s41586-019-1058-x>) and Pasolli et al.

(<https://doi.org/10.1016/j.cell.2019.01.001>) both utilized single-sample assemblies for metagenomic binning, yielding 60,664 and 154,723 MAGs respectively. Further, both manuscripts investigated the metagenomic binning differences from co-assemblies and single-sample assemblies. From Nayfach et al:

We performed single-sample assembly and binning (rather than co-assembly) to preserve strain variation between human hosts, and because co-assembly was not computationally feasible for our large dataset. On the basis of a subset of samples, our pipeline produced 1.8× more non-redundant high-quality MAGs compared to co-assembly, and 3.3× more than a previous study²⁰ that used abundance co-variation across samples (Extended Data Fig. 2).

This comparison demonstrates that metagenomic binning from single-sample assemblies compares favorably, if not better, than binning results from co-assemblies. These results from the human gut metagenome should be applicable to rumen metagenomes as well. Consequently, the literature supports the use of single-sample assemblies for metagenomic binning.

The authors are correct that my second comment contained a typo. I was indeed interested in knowing whether MAGs can be used for valid comparisons with other genomes. Thanks for the response, and the links to previous literature on this.

Thanks for the response.